# *Aspergillus fumigatus* calcium-responsive transcription factors regulate cell wall architecture promoting stress tolerance, virulence and caspofungin resistance

**Patrícia Alves de Castro**[1☉], **Ana Cristina Colabardini**[1☉], **Adriana Oliveira Manfiolli**[1], **Jéssica Chiaratto**[1], **Lilian Pereira Silva**[1], **Eliciane Cevolani Mattos**[1], **Giuseppe Palmisano**[2], **Fausto Almeida**[3], **Gabriela Felix Persinoti**[4], **Laure Nicolas Annick Ries**[3], **Laura Mellado**[1], **Marina Campos Rocha**[5], **Michael Bromley**[6], **Roberto Nascimento Silva**[3], **Gabriel Scalini de Souza**[7], **Flávio Vieira Loures**[7], **Iran Malavazi**[5], **Neil Andrew Brown**[8]*, **Gustavo H. Goldman**[1]*

1 Departamento de Ciências Farmacêuticas, Faculdade de Ciências Farmacêuticas de Ribeirão Preto, Universidade de São Paulo, Ribeirão Preto, Brazil, 2 Departamento de Microbiologia, Instituto de Ciências Biomédicas, Universidade de São Paulo, São Paulo, Brazil, 3 Departamento de Bioquímica e Imunologia, Faculdade de Medicina de Ribeirão Preto, Universidade de São Paulo, Ribeirão Preto, Brazil, 4 Laboratório Nacional de Biorrenováveis (LNBR), Centro Nacional de Pesquisa em Energia e Materiais (CNPEM), Campinas, São Paulo, Brasil, 5 Departamento de Genética e Evolução, Centro de Ciências Biológicas e da Saúde, Universidade Federal de São Carlos, São Carlos, São Paulo, Brazil, 6 Manchester Fungal Infection Group, Institute of Inflammation and Repair, University of Manchester, Manchester, United Kingdom, 7 Instituto de Ciência e Tecnologia, Universidade Federal de São Paulo, São José dos Campos, Brazil, 8 Department of Biology & Biochemistry, University of Bath, Claverton Down, Bath, United Kingdom

☉ These authors contributed equally to this work.
* N.A.Brown@bath.ac.uk (NAB); ggoldman@usp.br (GHG)

**Data Availability Statement:** RNAseq data for the wild-type, Δ*crzA*, and Δ*zipD* are available from

## Abstract

*Aspergillus fumigatus* causes invasive aspergillosis, the most common life-threatening fungal disease of immuno-compromised humans. The treatment of disseminated infections with antifungal drugs, including echinocandin cell wall biosynthesis inhibitors, is increasingly challenging due to the rise of drug-resistant pathogens. The fungal calcium responsive calcineurin-CrzA pathway influences cell morphology, cell wall composition, virulence, and echinocandin resistance. A screen of 395 *A. fumigatus* transcription factor mutants identified nine transcription factors important to calcium stress tolerance, including CrzA and ZipD. Here, comparative transcriptomics revealed CrzA and ZipD regulated the expression of shared and unique gene networks, suggesting they participate in both converged and distinct stress response mechanisms. CrzA and ZipD additively promoted calcium stress tolerance. However, ZipD also regulated cell wall organization, osmotic stress tolerance and echinocandin resistance. The absence of ZipD in *A. fumigatus* caused a significant virulence reduction in immunodeficient and immunocompetent mice. The Δ*zipD* mutant displayed altered cell wall organization and composition, while being more susceptible to macrophage killing and eliciting an increased pro-inflammatory cytokine response. A higher number of neutrophils, macrophages and activated macrophages were found in Δ*zipD* infected mice lungs. Collectively, this shows that ZipD-mediated regulation of the fungal cell wall

NCBI's Short Read Archive under the Bioproject (PRJNA445394).

**Funding:** São Paulo Research Foundation (FAPESP) grant numbers 2016/07870-9 (GHG), 2016/12948-7 (PAC), 2014/24951 (AOM), 2016/22062-6 (JC), 2016/21392-2 (LPS), 2017/19288-5 (ECM), 2014/06863-3 (GP), 2014/04783-2 (FVL) and 2017/07536-4 (ACC) and Conselho Nacional de Desenvolvimento Científico e Tecnológico (CNPq) (GHG), both from Brazil. NB was supported by the BBSRC Future Leader Fellowship (BB/N011686/1). The funders had no role in study design, data collection and analysis, decision to publish, or preparation of the manuscript.

**Competing interests:** The authors have declared that no competing interests exist.

contributes to the evasion of pro-inflammatory responses and tolerance of echinocandin antifungals, and in turn promoting virulence and complicating treatment options.

## Author summary

*A. fumigatus* is the main ethiological agent of one of the most important life-threatening fungal diseases in immuno-compromised humans, invasive aspergillosis. Treatment commonly involves echinocandin antifungal drugs that inhibit cell wall β-1,3-glucan polysaccharide biosynthesis. Calcium is an important messenger for many signaling pathways regulating the fungal response to stress and antifungal drugs. The calcium responsive calcineurin phosphatase influences the localization and activity of the CrzA transcription factor (TF), regulating the activation of several stress responses and cell wall modifications. For many years, CrzA has been recognized as the sole calcium/calcineurin-dependent TF. Here, we identify nine TFs involved in the calcium/calcineurin metabolism, including a novel *A. fumigatus* calcium/calcineurin dependent TF named ZipD. Transcriptional profiling of the response of *A. fumigatus* wild-type, plus the ΔcrzA and ΔzipD mutant, strains shows CrzA and ZipD to have shared and unique functions. ZipD was found to be important for not only calcium metabolism, but also for the cell wall organization, osmotic stress and echinocandin tolerance. During host infection, ZipD plays an important role in modulating fungal cell walls, promoting evasion of the host pro-inflammatory immune responses and virulence. Our work emphasizes the complexity and importance of novel additional mechanisms for calcium signaling, including ZipD, to regulate fungal stress responses, cell wall structure, promoting pathogenesis and antifungal resistance, complicating disease control.

## Introduction

*Aspergillus fumigatus* is a filamentous fungus that can cause disease in humans [1]. Depending on a patient's immunological status, *A. fumigatus* can cause a distinct set of clinical disorders that extend from severe allergies to lethal disseminated infections [1]. Invasive aspergillosis (IA) is the most common life-threatening fungal diseases in immuno-compromised humans and mortality rates can reach 90% [2–6]. Disseminated fungal infections are treated with antifungal drugs, including polyenes, azoles, and echinocandins [7]. However, infections by antifungal drug-resistant pathogens are on the rise, presenting severe treatment constraints [8].

Sensing and withstanding the host environment is essential for *A. fumigatus* virulence [8, 9]. Calcium is an essential secondary messenger and modulates the conformation of calcium-binding proteins, such as calmodulin, which activates calmodulin-dependent enzymes including the calcineurin phosphatase [10, 11]. In multiple fungal species, calcineurin regulates the activity of the Crz1p/CrzA transcription factor [12–15] with the phosphorylated form accumulating in the cell cytosol. In response to certain stimuli, which increase cytosolic calcium, calcineurin dephosphorylates Crz1p/CrzA leading to its re-localization to the nucleus. Crz1p/CrzA contains a $C_2H_2$ zinc finger motif that binds to a specific CDRE (<u>c</u>alcineurin-<u>d</u>ependent <u>r</u>esponse <u>e</u>lement) at gene promoters, to direct calcium- and calcineurin-dependent gene expression [16].

In *A. fumigatus* calcineurin is not essential for viability, but regulates differentiation, morphology, fitness, and virulence [17, 18]. In response to stimuli that increase cytosolic calcium,

calcineurin regulates the nuclear localization and activity of the CrzA transcription factor [12–15]. Furthermore, calcineurin-CrzA pathway promotes antifungal resistance, since combining calcium or calcineurin inhibitors with azoles or echinocandins restores drug-mediated growth inhibition for some antifungal resistant fungal strains [16, 19–25]. In *A. fumigatus* CrzA mediates cellular tolerance to increased concentrations of calcium and manganese, controls the transcription of several calcium transporters and channels, and is important for virulence [26–32].

Echinocandins, such as caspofungin, are non-competitive β-1,3-glucan synthase inhibitors, which impair fungal cell wall polysaccharide biosynthesis and integrity [33,34]. High caspofungin concentrations revert the anticipated inhibition of *A. fumigatus* growth, a phenomenon termed the caspofungin paradoxical effect (CPE) [3]. High caspofungin concentrations elicit a spike in cytosolic calcium, which activates calcineurin and CrzA to promote the CPE [35, 36]. Additionally, calcineurin is involved in regulating the expression of chitin synthases, providing a link between the mitogen-activated cell wall integrity (CWI) and calcium signaling pathways in the regulation of fungal cell wall biosynthesis [32, 35, 37]. CrzA influences cell wall organization in the absence of caspofungin, but also regulates the expression of specific chitin synthases during the CPE [32].

Previously, a novel basic leucine zipper ZipD transcription factor was identified to function in the calcium-calcineurin pathway and was involved in the CPE [24]. Additional transcription factors may therefore participate in calcium signaling. A screen of 395 *A. fumigatus* transcription factor mutants identified nine important to calcium stress tolerance, including CrzA and ZipD. Here, comparative transcriptomics revealed CrzA and ZipD regulated converged and distinct gene expression networks and cellular stress responses. CrzA and ZipD additively promoted calcium stress tolerance. However, ZipD additionally regulated fungal cell wall organization and osmotic stress tolerance. Consequently, during infection ZipD contributed to evading immune activation by regulating its cell wall composition and organization and preventing the exposure of immune stimulatory molecules within the host, while promoting virulence and echinocandin resistance. This again shows the complexity and importance of calcium signaling to fungal pathogenesis and disease control.

## Results

### Nine *A. fumigatus* transcriptional factors govern calcium tolerance

The transcriptional factors (TFs), CrzA and ZipD, have roles in calcium stress tolerance. To assess if any other TFs play a role in this stress response, a library of 395 *A. fumigatus* TF null mutants (S1 Table) was screened for sensitivity to high concentrations of calcium. Seven null mutants exhibited increased sensitivity to 500 mM calcium [ΔAfu1g10550, ΔAfu5g10620, ΔAfu6g12522 (*skn7*), ΔAfu4g07090, ΔAfu7g03910 (*nsdC*), and as previously described ΔAfu2g03280 (*zipD*) ΔAfu1g06900 (*crzA*)] [26–32], while two mutants showed increased tolerance [ΔAfu1g13190 and ΔAfu3g08010 (*ace1*)] (Fig 1A). The calcium-dependent phosphatase calcineurin is a key regulator of CrzA, dephosphorylating the TF, prior to its nuclear import. To identify if any of the TFs identified in the screen are associated with calcineurin activity, the null mutants were screened for increased sensitivity to the calcineurin inhibitor cyclosporin. Three of the null mutants [Afu2g03280 (*zipD*), Afu5g10620, and Afu6g12522 (*skn7*)] that were sensitive to high calcium concentrations were also hypersensitive to cyclosporin. The two mutants that had increased calcium tolerance were also more resistant to cyclosporin (Fig 1B). Interestingly, as shown previously, Δ*crzA* displays the same sensitivity to cyclosporin as the wild-type strain (Fig 1B) [27, 32].

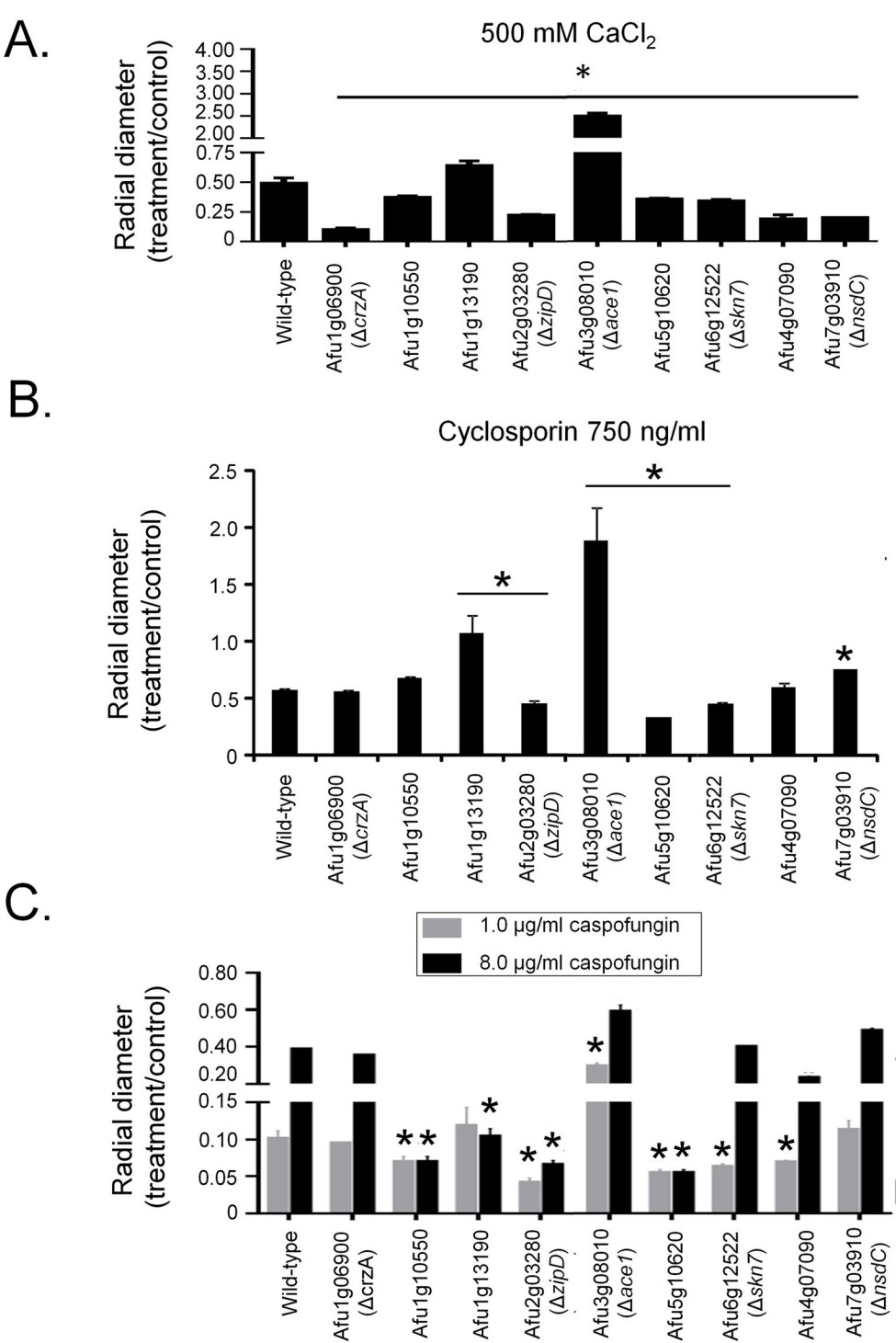

**Fig 1. Identification of *A. fumigatus* calcium-sensitive mutants.** A collection of *A. fumigatus* 395 null transcription factor mutants was screened for sensitivity to calcium by growing them on minimal medium+500 mM $CaCl_2$. Nine mutants that displayed altered sensitivity or an improvement in their growth in the presence of $CaCl_2$ had their radial growth quantified on 500 mM $CaCl_2$ (A), 750 ng cyclosporin (B) or 1 or 8 μg/ml of caspofungin (C). The results are the means of three repetitions ± standard deviation. Statistical analysis was performed using a one-tailed, paired t-test when compared to the control condition (*, $p < 0.005$).

The calcium-calcineurin pathway is directly associated with caspofungin resistance and the CPE [3]. Therefore, these TFs may also have roles in the CPE. Caspofungin inhibited growth of the *A*. *fumigatus* wild-type strain by 90% at 1.0 μg/ml, but due to CPE only by 60% at 8.0 μg/ml (Fig 1C). The null mutants corresponding to the genes Afu1g10550, Afu2g03280 (*zipD*), Afu5g10620, Afu6g12522 (*skn7*), and Afu4g07090 were more sensitive than the wild-type strain to caspofungin at 1.0 μg/ml and exhibited a reduction in the CPE at 8.0 μg/ml (Fig 1C). In contrast, the mutant that lacks Afu3g08010 (*ace1*), which was more resistant to calcium stress and cyclosporin, was also significantly more resistant to caspofungin at 1.0 μg/ml (Fig 1C). Taken together, these results strongly suggest that in addition to CrzA, additional transcription factors, including ZipD and Afu5g10620, are involved in the *A*. *fumigatus* calcium-calcineurin pathway and contribute to the tolerance and the CPE.

## ZipD and CrzA mediated pathways genetically interact during calcium stress

CrzA is described to be the primary TF activated in a calcineurin-dependent manner upon exposure to high calcium concentrations and in response to several other stresses [14]. The calcineurin inhibitor cyclosporin blocks the nuclear import of ZipD, implicating its involvement in the calcineurin pathway. Phylogenetic analysis of ZipD across 791 fungi, representing the 13 different taxonomic classes or subphyla within Dikarya revealed that orthologues were restricted to Pezizomycotina, in particular Eurotiomycetes (S1A Fig). ZipD orthologues were identified in other important fungal pathogens, such as *A*. *flavus*, *Penicillium spp*, *Coccidioides spp*, *Talaromyces stipitatus*, and *Ajellomyces capsulatus* (S1B Fig).

The Δ*zipD* mutants and Δ*crzA* were more sensitive to calcium than the corresponding wild-type and complemented strains (Fig 2A) [32]. The relationship between these two calcium-responsive TFs, ZipD and CrzA, was assessed by constructing the double Δ*crzA* Δ*zipD* mutant, which was more sensitive to calcium than the corresponding single mutants (Fig 2A). This implies that ZipD and CrzA genetically interact, where calcium stress promotes either a single pathway in the bifurcated activation, or parallel signaling pathways in activating, the two TFs to regulate the calcium stress response.

The transcriptional relationship between *zipD* and *crzA* was assessed by RT-qPCR during calcium stress. The *cofA* (Afu5g10570) gene, which encodes for the actin-binding protein cofilin, was used as a normalizer due to its consistent expression in all strains during calcium stress (S2 Table). A putative vacuolar $H^+/Ca^{2+}$ exchanger encoded by *vcxB* (Afu2g07630) was used as a positive control for calcium stress induction [27]. Both ZipD and CrzA modulated the expression of the calcium-responsive gene *vcxB* upon calcium stress, showing their importance to regulating the transcriptional calcium stress response (Fig 2B). There is no mRNA accumulation of the *vcxB* gene in the Δ*crzA* Δ*zipD* mutant (see Fig 2B). However, *zipD* was not induced upon calcium stress, in either the wild-type or Δ*crzA* strains (Fig 2C). Similarly, *crzA* was not induced in the wild-type strain upon calcium stress (Fig 2D). This contrasts Soriani *et al*. (2008) [28] which cite an increase in *crzA* expression upon calcium stress. This difference is explained by the use of the *tubA* (Afu1g10910) normalizer, which here exhibited reduced expression in the calcium stressed wild-type strain (S3 and S4 Tables). However, a 40% and a 6-fold increase in *crzA* accumulation were observed in the Δ*zipD* mutant, in comparison with the wild-type strain when exposed to 10 and 30 mins calcium stress respectively (Fig 2D). This shows that *zipD* and *crzA* are not transcriptionally induced by calcium stress, but *crzA* mRNA could accumulate in response to calcium signaling in the absence of *zipD*, potentially reflecting a compensatory mechanism.

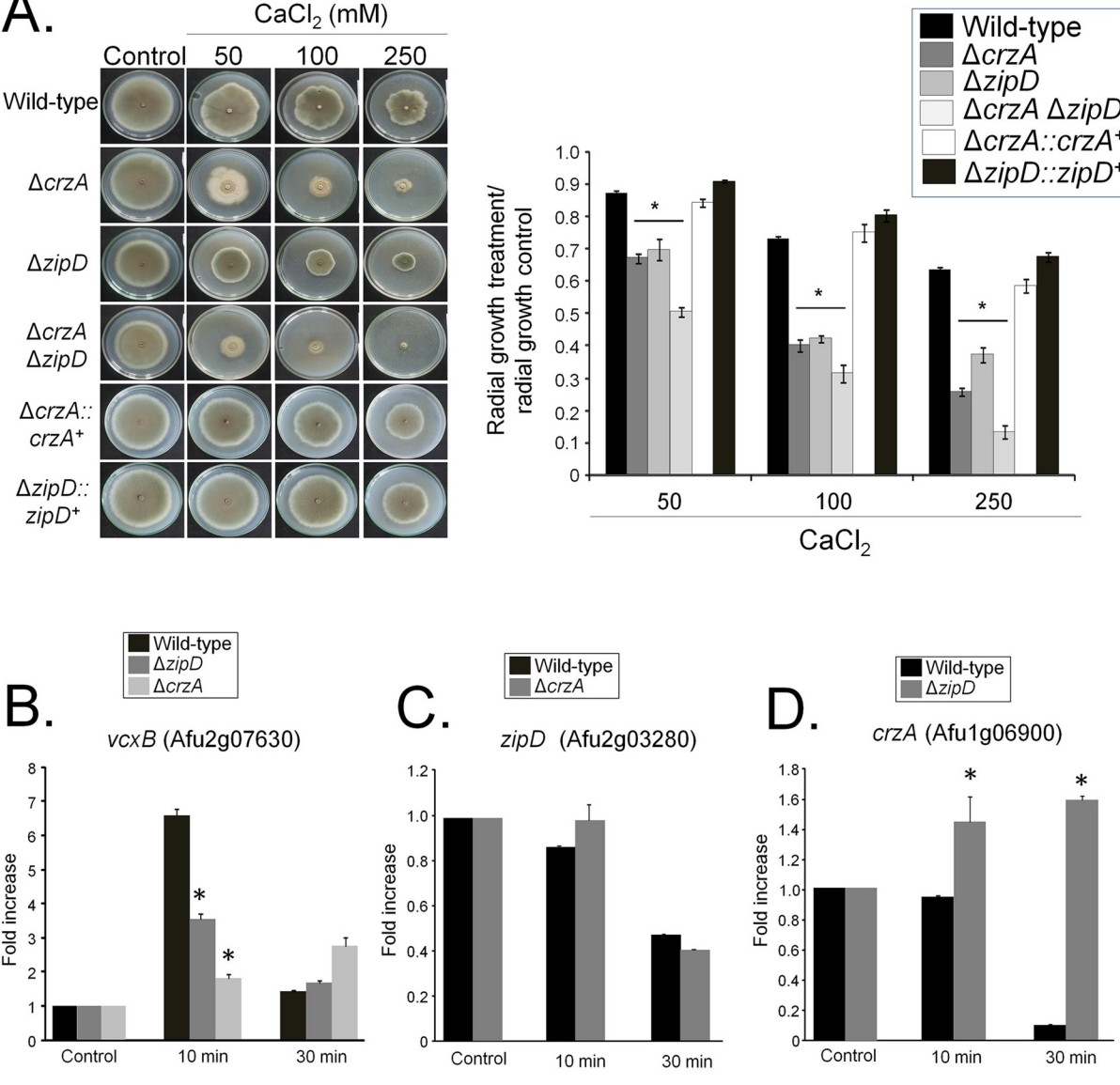

**Fig 2. A genetic interaction between ΔcrzA and ΔzipD.** (A) The wild-type, ΔcrzA, ΔzipD, and ΔcrzA ΔzipD mutant strains were grown on minimal media with increasing concentrations of CaCl₂ for 5 days at 37°C. The results are expressed as the average of three repetitions ± standard deviation. (B to D) Determination of the expression of *vcxB*, *zipD*, and *crzA* by qRT-PCR. The wild-type, ΔzipD, and ΔcrzA strains were grown for 16 h at 37°C and transferred to 200 mM CaCl₂ for 0, 10, and 30 mins. Gene expression was normalized using *cofA* (Afu5g10570). Standard deviations present the average of three independent biological repetitions (each with 2 technical repetitions). Statistical analysis was performed using a one-way ANOVA test when compared to the wild-type condition (*p<0.05).

### ZipD and CrzA share common and unique transcriptional signatures

To identify ZipD-dependent and CrzA-dependent gene expression, the wild-type and ΔzipD and ΔcrzA transcriptomes were assessed post exposure to 200 mM calcium for 10 and 30 mins (Fig 3A). In 200 and 250 mM there was about 50 and 70% growth inhibition of the wild-type and mutant strains, respectively (Fig 2A), and these two timepoints will allow us to identify early and late genes induced by calcium. Differentially expressed genes were defined as those with a minimum of two-fold change in gene expression (log2FC ≥ 1.0 and ≤ -1.0; FDR of 0.05) when compared to the wild-type strain under the equivalent conditions. Venn analyses

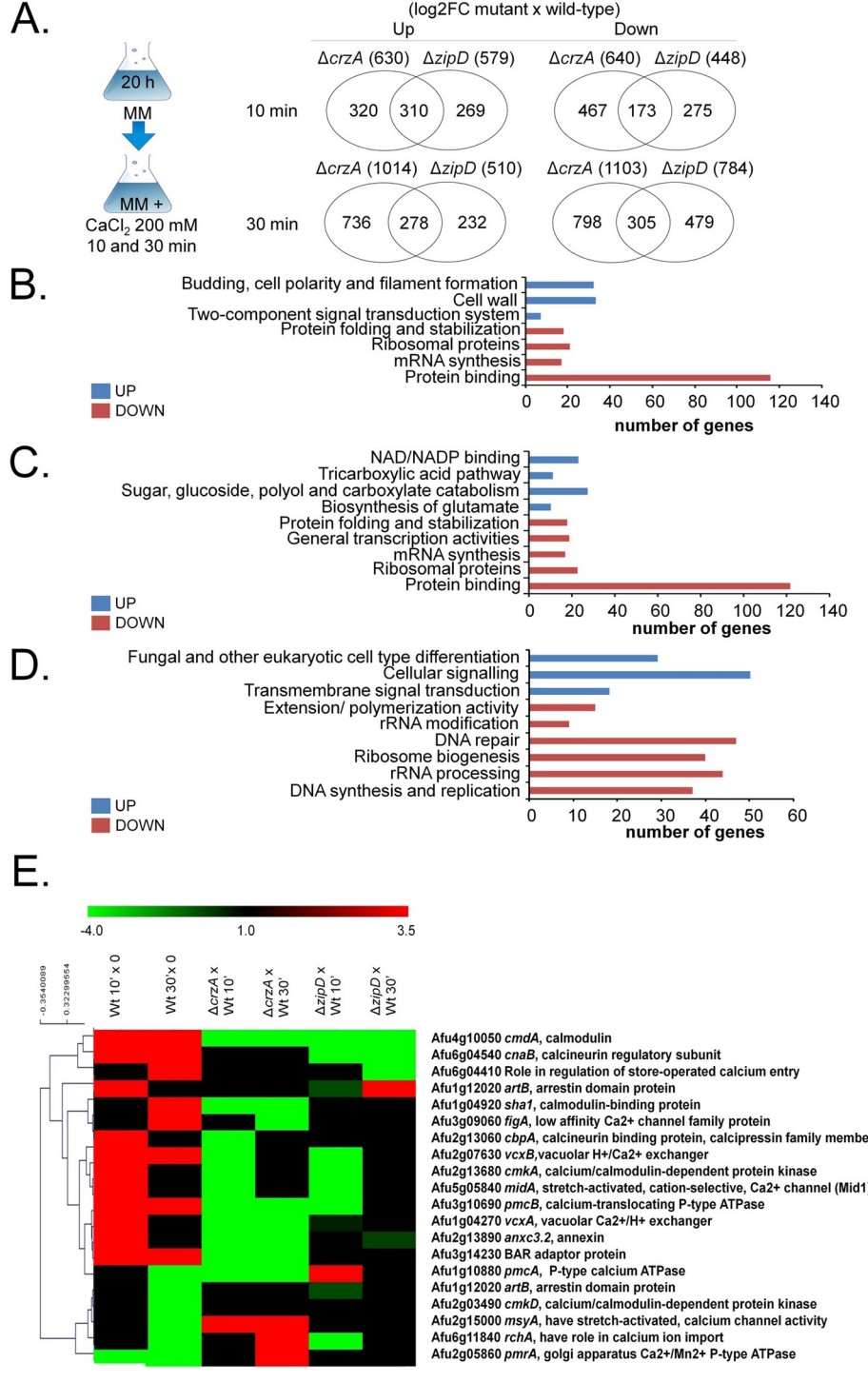

**Fig 3. The ΔcrzA and ΔzipD show common and distinct transcriptional profiles.** (A) Venn diagram comparing the genes specifically expressed in each mutant relative to the wild-type strain. (B) A summary of the FunCat terms over-represented up or down regulated (adjusted p-value < 0.05) in log2FC ΔzipD versus ΔcrzA post transfer to 200 mM CaCl₂ for 10 and 30 mins. For the full list refer to S3–S8 Tables. (C) A summary of the FunCat terms over-represented up or down regulated (adjusted p-value < 0.05) but uniquely expressed in ΔzipD in log2FC ΔzipD versus wild-type post transfer to 200 mM CaCl₂ for 10 and 30 mins. For the full list refer to S2–S8 Tables. (D) A summary of the FunCat terms over-represented up or down regulated (adjusted p-value < 0.05) but uniquely expressed in ΔcrzA in log2FC ΔzipD versus wild-type post transfer to 200 mM CaCl₂ for 10 and 30 mins. For the full list refer to S2–S8 Tables. (E) The RNA-seq data for annotated genes corresponding to the expression of genes encoding selected proteins putatively

involved in calcium metabolism. The wild-type is shown as 10 and 30 min calcium stress versus time zero (20 hours growth), and gene deletion strains are shown as the deletion strain versus the equivalent wild-type 10 and 30 min time points (the mutant values have been normalised to the basal level of expression of each gene before stress, i.e., expression ratios are being compared: wild-type 10 min versus time zero divided by a specific mutant 10 and 30 min versus time zero). Hierarchical clustering was performed in MeV (http://mev.tm4.org/), using Pearson correlation with complete linkage clustering.

of the genes differentially expressed in Δ*zipD* and Δ*crzA* versus the wild-type strain at either 10 or 30 mins exposure to calcium stress (S3–S8 Tables) revealed a complex pattern of shared and mutant-specific transcriptional responses to calcium stress (Fig 3A). The mutants shared 516 (total 588 in both 10 and 30 min) and 462 (total 478 in both 10 and 30 min) unique up- or down-regulated genes, respectively (Fig 3A). Mutant-specific transcriptional responses were more abundant, where 862 (total 1,056 in both 10 and 30 min) and 1,053 (total 1,265 in both 10 and 30 min) unique genes were up- or down-regulated for Δ*crzA*, and 457 (total 501 in both 10 and 30 min) and 710 (total 754 in both 10 and 30 min) unique genes were up- or down-regulated for Δ*zipD* (Fig 3A).

FunCat enrichment analyses revealed the distinct biological functions of the shared and mutant-specific transcriptional differences. The Δ*zipD* and Δ*crzA* mutants shared 516 up-regulated genes involved in budding, cell polarity and filament formation, cell wall, and two-component signal transduction system (sensor kinase component) (Fig 3B). The 462 genes down-regulated in both Δ*zipD* and Δ*crzA* encoded for proteins involved in protein folding and stabilization, ribosomal proteins, mRNA synthesis and protein binding (Fig 3B). The 457 genes specifically up-regulated in Δ*zipD* showed the induction of genes involved in the biosynthesis of glutamate, sugar, glucoside, polyol, and carboxylate catabolism (Fig 3C). The 862 genes specifically up-regulated in Δ*crzA* showed the induction of genes involved in the regulation of C-compound and carbohydrate metabolism, transmembrane signal transduction, cellular signaling, and fungal differentiation (Fig 3D). Finally, the 710 and 1,053 genes specifically down-regulated in either the Δ*zipD* or Δ*crzA* strains encoded for distinct proteins which were similarly involved in mRNA synthesis, ribosomes, and protein folding and stabilization (Fig 3C and 3D). Taken together, these results indicate that both ZipD and CrzA influence fungal metabolism, nucleic acid and protein biosynthesis, replication, differentiation and cell signaling in a shared and distinct manner.

The FunCat analysis also indicated 20 genes encoding for proteins involved in calcium metabolism, such as calcium transporters, channels, and protein kinases, were differentially modulated (Fig 3E). Several genes were dependent on both TFs, such as *cmdA* (Afu4g10050) encoding calmodulin, whereas ZipD and CrzA specifically modulated other genes (Fig 3E) including *cbpA* (Afu2g13060, calcineurin-binding protein) and *cnaB* (Afu6g04540, calcineurin regulatory subunit).

## ZipD and CrzA mediated calcium signaling impacts on transcriptional regulation of osmotic stress and cell wall biosynthesis pathways

The analysis of genes concomitantly up-regulated in both the Δ*zipD* and Δ*crzA* mutants upon calcium stress revealed several components of the *A. fumigatus* HOG (High Osmolarity Glycerol) pathway to have elevated expression profiles (Fig 4A and S3–S8 Tables). This included the *shoA* (Afu5g08420) transmembrane osmosensor, the *fhk6* (Afu4g00320) histidine kinase, *tcsA* (Afu6g10240) histidine kinase, two-component signal transduction protein [38], *phkA* (Afu3g12550) histidine-containing phosphotransfer protein [39], response regulators *ssk1* (Afu5g08390) [39], *srrB* (Afu2g15010) [39], *fhk2* (Afu3g07130), and *flk5* (Afu4g00660) [40], the trancription factor *sebA* (Afu4g09080) involved in osmotic and oxidative stresses [29] plus

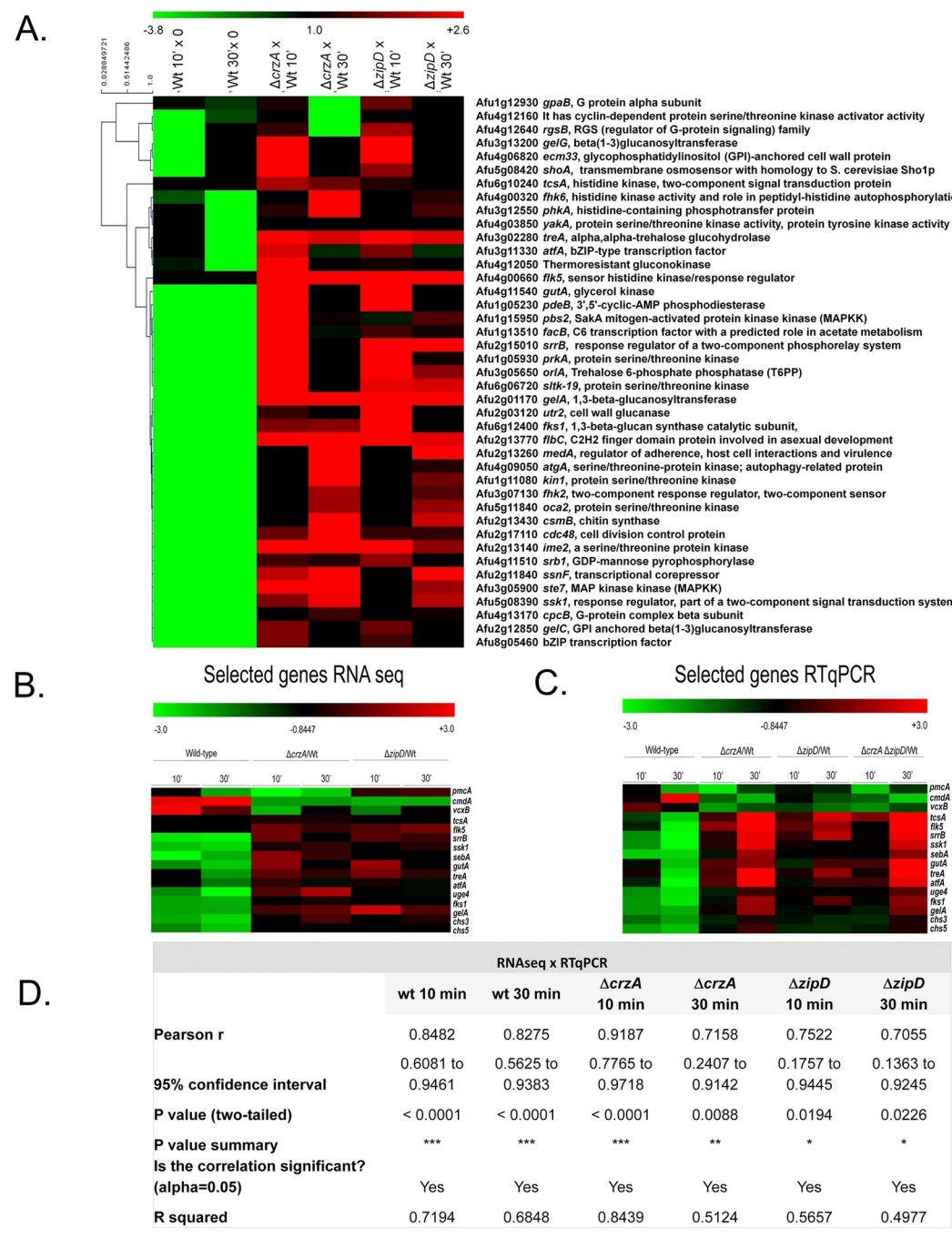

**Fig 4. The Δ*crzA* and Δ*zipD* influence calcium, osmotic stress and signaling, and cell wall metabolism.** The RNA-seq data for annotated genes corresponding to the expression of genes encoding selected proteins putatively involved in calcium metabolism (A) Selected proteins putatively involved in calcium signal transduction, osmotic stress, cell wall biosynthesis, and transcription factors.The wild-type is shown as 10 and 30 min calcium stress versus time zero (20 hours growth), and gene deletion strains are shown as the deletion strain versus the equivalent wild-type 10 and 30 min time points (the mutant values have been normalised to the basal level of expression of each gene before stress, i.e., expression ratios are being compared: wild-type 10 min versus time zero divided by a specific mutant 10 and 30 min versus time zero). Hierarchical clustering was performed in MeV (http://mev.tm4.org/), using Pearson correlation with complete linkage clustering. (B) Heatmap for the RNAseq values of sixteen selected genes involved in calcium metabolism, osmotic stress, and cell wall metabolism. (C) Heat map of RTqPCR for the same sixteen selected genes from (B). The wild-type, Δ*crzA*, Δ*zipD*, and Δ*crzA* Δ*zipD* mutant strains were grown for 20 h at 37˚C and transferred to 200 mM CaCl₂ for 0, 10, and 30 mins. Gene expression was normalized using *cofA* (Afu5g10570). Standard deviations present the average of three independent biological repetitions (each with 2 technical repetitions). (D) The expression of these sixteen genes as measured by RT-qPCR showed a high level of correlation with the RNA-seq data (Pearson correlation from 0.7055 to 0.9187).

the *pbs2* (Afu1g15950) mitogen-activated protein kinase, and the HOG1/SakA (Afu1g12940) mitogen-activated protein kinase (Fig 4A). Genes important for the osmotic stress response, such as the *treA* (Afu3g02280), alpha, alpha-trehalose glucohydrolase, the *orlA* trehalose 6-phosphate phosphatase, the *gutA* (Afu4g11540) glycerol kinase, and the main *atfA* bZIP-type TF, were also identified [41] (Fig 4A and S3–S8 Tables).

We also observed genes up-regulated in both mutants that are involved in the biosynthesis and/or remodeling of the cell wall, such as *fks1* (Afu6g12400) 1,3-β-glucan synthase catalytic subunit, major subunit of glucan synthase [42], *uge4* (Afu4g14090) UDP-glucose 4-epimerase [43], *gelA* (Afu2g01170) 1,3-β-glucanosyltransferase with a role in elongation of 1,3-beta-glucan chains [44], *chs3* (Afu3g05580) encoding a protein with a predicted role in the regulation of chitin synthase activity [45], and *chs5* (Afu6g02510) chitin biosynthesis protein (Fig 4A and S3–S8 Tables).

### Independent validation that ZipD and CrzA influence transcriptional regulation of calcium, cell wall and osmotic stress responses

RT-qPCR experiments validated the RNA-seq results for the majority of the 16 selected genes from the calcium stress, osmotic and cell wall stress responses (Figs 4B and 4C and S2). Additionally, the majority of these 16 genes showed the same dysregulated transcription in the Δ*crzA* Δ*zipD* double mutant, as previously observed in the either one of the single Δ*zipD* or Δ*crzA* mutants (Figs 4B and 4C and S2). The expression of these 16 genes showed a high level of correlation with the RNA-seq data (Pearson correlation from 0.7055 to 0.9187; Fig 4D). These results show that ZipD and CrzA have shared and TF-specific functions in modulating the response to calcium stress, affecting directly or indirectly the expression of genes involved in osmotic stress and cell wall biosynthesis and/or remodeling.

### ZipD and CrzA promote cell wall and osmotic stress tolerance through their regulation of cell wall composition and architecture

Transcriptional profiling suggests that ZipD plays a role in calcium metabolism, plus the cell wall and osmotic stress responses. In addition to Δ*zipD* and Δ*crzA* Δ*zipD* mutant strains exhibiting increased sensitivity to caspofungin and an absence of the CPE [32], they were also more sensitive to other cell wall damaging agents, including congo red (CR) and calcofluor white (CFW), when compared to the wild-type, Δ*crzA*, and complemented strains (Fig 5A–5C). Increased sensitivity to cell wall stressors suggested ZipD influenced cell wall composition. All mutants have different β-1,3-glucan levels in their cell walls than the wild-type strain (Fig 5D); however, Δ*crzA* has lower levels than the Δ*zipD* and Δ*crzA* Δ*zipD* strains, but Δ*crzA* is different from the double mutant ($p < 0.001$; Fig 5D). The three mutants also have higher chitin levels in their cell walls than the wild-type strain (Fig 5E); however, Δ*crzA* has higher levels than the Δ*zipD*, but Δ*zipD* is different from the double mutant ($p < 0.001$; Fig 5E). Therefore, while ZipD and not CrzA plays a major role in the cell wall stress response, TFs similarly influenced cell wall composition.

Subsequently, differences in cell wall organization were assessed by determining the exposure of different polysaccharides on the cell surface. Dectin-1 binding only revealed the reduced exposure of β-glucans in the Δ*crzA* cell walls when compared with the wild-type, Δ*zipD* and complemented strains (Fig 5F). However, CFW staining showed all the Δ*zipD*, Δ*crzA* and Δ*crzA* Δ*zipD* mutants had a comparable reduction in chitin exposure compared to the wild-type and complemented strains (Fig 5G). Transmission electron microscopy (TEM) showed the cell wall of all three mutants was thicker than the wild-type strain (Fig 5H and S9 Table); where Δ*zipD* and Δ*crzA* Δ*zipD* mutants showed a greater increase in cell wall thickness

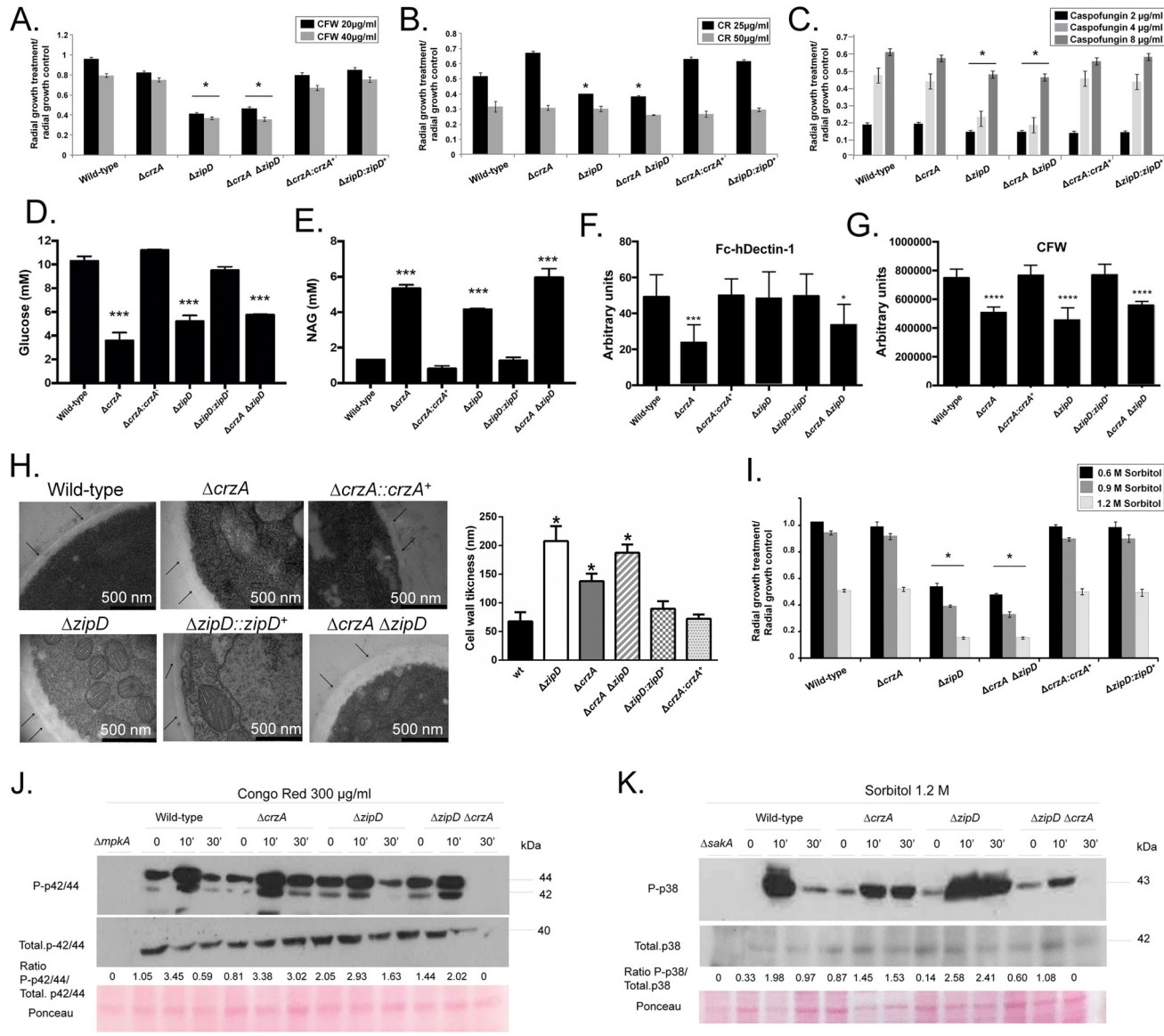

**Fig 5. The ΔzipD mutant strain is more sensitive to cell wall damaging agents.** (A) The wild-type, the mutant, and the complemented strains were grown on minimal media (MM) with increasing concentrations of (A) calcofluor white (CFW), (B) congo red (CR), and (C) caspofungin for 5 days at 37°C. The results are expressed as the average of three repetitions ± standard deviation. (D-E) Glucose and N-acetyl glucosamine (NAG) concentrations, as determined by high performance liquid chromatography (HPLC), in mycelial extracts of the *A. fumigatus* strains when grown for 16 h in MM at 37°C. (F-G) Detection of the β-1,3–glucan and chitin exposed on the cell surface. Conidia were cultured in liquid MM to the hyphal stage, UV-killed, and stained with CFW or soluble dectin-1 to detect the content of exposed chitin or β-glucan. Experiments were performed in triplicate and displayed as mean values with standard errors (Two-way Anova followed by Tukey's $p < 0.05$) (H) Transmission electron microscopy of mycelial sections of the *A. fumigatus* strains when grown for 24 h in MM at 37°C. Arrows indicate the external borders of the cell wall. The cell wall thickness (nm) of 100 sections of different hyphal germlings (average of 4 sections per germling) were measured when grown in the same conditions as specified under (H). Standard deviations present the average of the 50 measurements and statistical analysis was performed using a one-tailed, paired t-test when compared to the control condition (*, $p < 0.00001$). (I) The wild-type, ΔcrzA, ΔzipD, ΔcrzA ΔzipD, and the complemented strains were grown on minimal media with increasing concentrations of sorbitol for 5 days at 37°C. The results are expressed as the average of three repetitions ± standard deviation. Asterisk indicates significant differences between different lineages (Two-way Anova followed by Tukey's $p < 0.001$). (J and K) Western blotting assays of SakA (K) and MpkA (J) phosphorylation in response to 1.2 M Sorbitol and Congo Red 300 μg/ml. Anti-P-p38 SakA, anti-p38 SakA, anti-P-p44/42 MpkA or anti-44/42 MpkA antibodies were used to detect the phosphorylation of SakA, total SakA, MpkA, and total MpkA, respectively. Signal intensities were quantified using the ImageJ software, and ratios of (P)-SakA to SakA or (P)-MpkA to MpkA were calculated. A Ponceau gel of the transfer protein filter served as an additional loading control.

than the Δ*crzA* strain (Fig 5H). Therefore, both ZipD and CrzA influence fungal cell wall composition and architecture, and have a complex interaction behavior. The Δ*zipD* and Δ*crzA* Δ*zipD* mutant strains were also more sensitive to osmotic stress (Fig 5I) than the wild-type, Δ*crzA*, and complemented strains, suggesting that the regulation of cell wall organization by ZipD also influenced osmotic stress tolerance.

The additive impact of the double Δ*crzA* Δ*zipD* mutant on stress tolerance was specific to calcium exposure. In comparison to the wild-type strain, the double mutant showed increased sensitivity to cell wall damaging agents, CFW, CR and caspofungin, osmotic stress, and altered cell wall organization (Fig 5A–5I). But these cell wall and osmotic stress phenotypes revealed no additive interaction, as stress tolerance in the double mutant was mostly reminiscent of Δ*zipD*.

## Absence of ZipD hyperactivates MpkA cell wall integrity and SakA osmotic stress MAPK signaling

To determine if the absence of ZipD or CrzA influenced the activation of the MpkA and SakA MAPK cascades in response to cell wall and osmotic stress, the phosphorylation levels of MpkA and SakA were monitored by immunoblot analysis. As expected MpkA and SakA phosphorylation was absent in the Δ*mpkA* and Δ*sakA* mutants (Fig 5J and 5K). In the wild-type strain, upon CR-induced cell wall stress, the MpkA phosphorylation increased 3-fold in 10 min, and recovered to basal levels after 30 min (Fig 5J). The three mutants however showed different MpkA phosphorylation levels to the wild-type strain. The Δ*crzA* mutant showed a delay in the recovery of basal phosphorylation state after 30 min, while the Δ*zipD* had higher phosphorylation in unstressed conditions, and the double mutant had reduced MpkA phosphorylation (Fig 5J). Then exposure of the wild-type strain to Sorbitol-induced osmotic stress, SakA phosphorylation increased 6-fold in 10 min and recovered after 30 min (Fig 5K). In contrast, in the Δ*crzA* mutant again showed a delay in recovery with increased SakA phosphorylation after 30 min exposure to Sorbitol, while the Δ*zipD* mutant showed the greatest increase in SakA phosphorylation, about 18-fold, after 10 and 30 min exposure to Sorbitol (Fig 5K). The Δ*crzA* Δ*zipD* double mutant however showed decrease SakA phosphorylation post exposure to Sorbitol (Fig 5K). This indicates that either CrzA and ZipD influence MAPK cascade activation in response to osmotic and cell wall stresses, or that the absence of these TFs causes increased cell wall and osmotic stress sensitivity which results in altered MAPK activation. These results suggest that ZipD and CrzA operate in two interacting pathways important for calcium metabolism, which have differing functions during the cell wall damage and osmotic stress responses.

## ZipD dephosphorylation upon calcium stress is not mediated by calcineurin

Translocation of ZipD:GFP to the nucleus upon exposure to high calcium or caspofungin concentrations is calcineurin-dependent [32]. It is unknown if ZipD is directly dephosphorylated by calcineurin. We exposed ZipD:3xHA strain to 50 to 200 mM CaCl$_2$ but surprisingly we always observed protein degradation under these conditions. However, we were able to extract intact proteins when Zip:3xHA cultures were exposed to 10 mM CaCl$_2$. Mass spectrometry analysis of immunoprecipitated ZipD:3xHA (S3 Fig) revealed a 25% decrease in the ZipD total protein when exposed to calcium stress (Fig 6A). Three putative phosphorylated peptides were identified at the ZipD N-terminal region (MASRKPSASILVPR) (Fig 6A and 6B). The identified KPSASILVPR ZipD peptide showed a similar concentration to the total ZipD protein. However, the concentration of the KPSASILVPR and MASRKPSASILVPR ZipD peptides

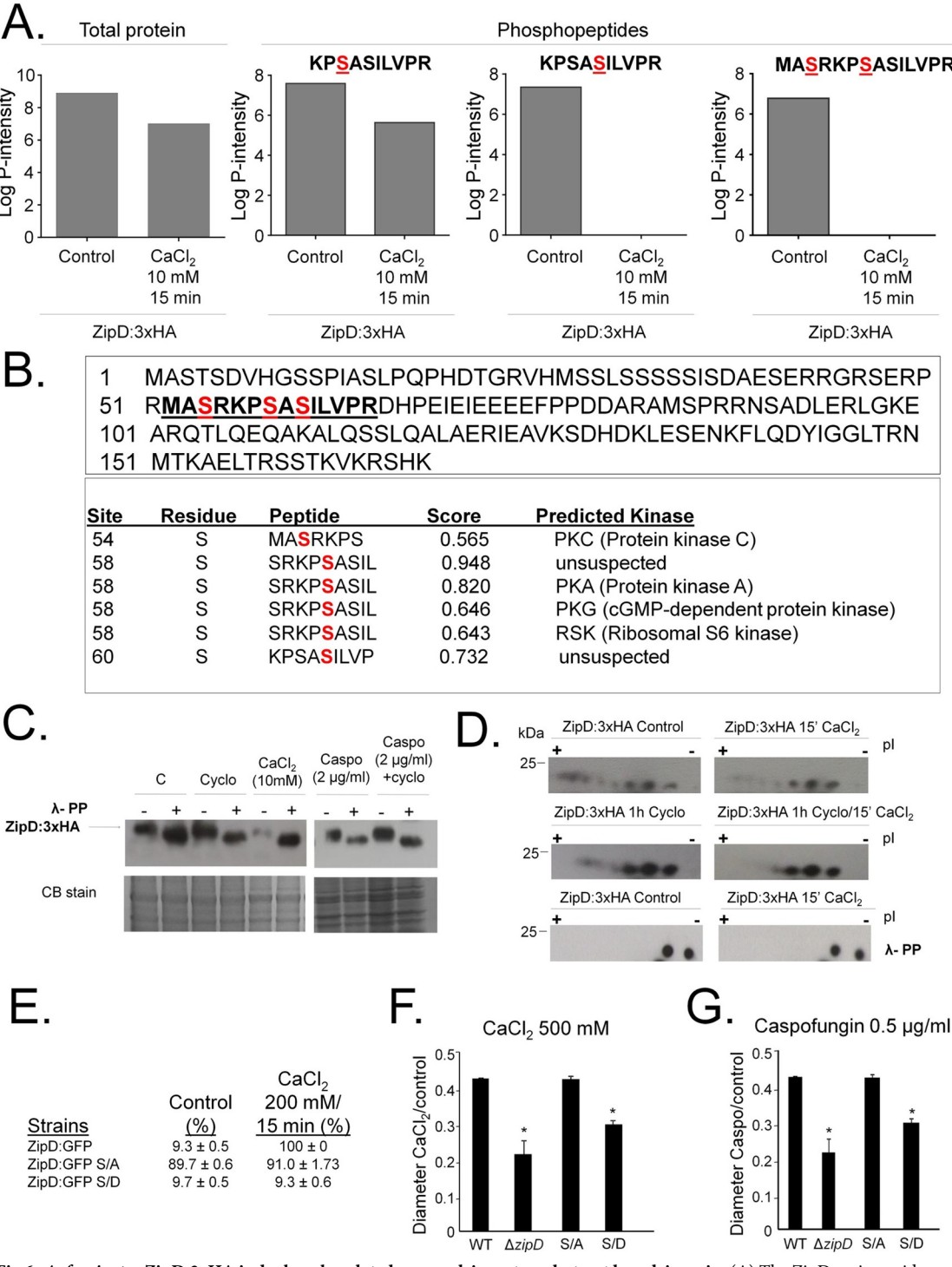

**Fig 6. *A. fumigatus* ZipD:3xHA is dephosphorylated upon calcium stress but not by calcineurin.** (A) The ZipD amino acid sequence. The underlined peptide shows the serine residues (in red) that were identified as phosphorylated. NetPhos 3.1 (http://www.cbs.dtu.dk/services/NetPhos/) predicted the serine residues to be phosphorylated by PKC (Protein kinase C), PKA (Protein kinase A), PKG (cGMP-dependent protein kinase), and RSK (Ribosomal S6 kinase). (B) Relative abundance of the identified phosphopeptides. (C) Immunoblot analysis for immunoprecipitated ZipD:3xHA. The *A. fumigatus* ZipD:3xHA strain was grown for 24 h at 37°C (C = Control) and transferred to either cyclosporin (Cyclo) 9 μg/ml for 1 h, CaCl₂ 10mM for 15 mins, or caspofungin 2.0 μg/ml for 1 h, or cyclosporin (9 μg/ml) for 1 h followed by 2.0 μg/ml caspofungin for another additional 1 h. Anti-HA antibody directed against the HA epitope was used to detect the ZipD:3xHA. The immunoprecipitated ZipD:3xHA was treated (+) or not (-) with Lambda phosphatase for 1 h at 30°C. A Coomassie Brilliant Blue (CBB)-stained gel is shown as an additional loading control.

(D) Immunoprecipitated ZipD:3xHA (same experimental design from C) was subjected to two-dimensional gel electrophoresis coupled with Western blotting. Anti-HA antibody directed against the HA epitope was used to detect the ZipD:3xHA. The immunoprecipitated ZipD:3xHA was treated (+) or not (-) with Lambda phosphatase for 1 h at 30˚C. (E) Percentage of ZipD:GFP translocated to the nuclei in ZipD:GFP, ZipD:GFP S/A, and ZipD:GFP S/D strains. (F and G) Growth phenotypes of ZipD:GFP, ZipD:GFP S/A, and ZipD:GFP S/D on $CaCl_2$ and caspofungin. The strains were grown for 5 days at 37˚C.

decreased when exposed to calcium stress (Fig 6A). The serine residues were predicted to be phosphorylated by PKC (Protein kinase C), PKA (Protein kinase A), PKG (cGMP-dependent protein kinase), or RSK (Ribosomal S6 kinase) by NetPhos 3.1 (http://www.cbs.dtu.dk/services/NetPhos/) (Fig 6B).

Immunoprecipitation of a functional ZipD:3xHA, post exposed to different concentrations of calcium or caspofungin, in the presence or absence of the calcineurin inhibitor cyclosporin, suggested that ZipD was not dephosphorylated by calcineurin (Fig 6C). In response to calcium or caspofungin treatments, the molecular weight of the single ZipD:3xHA band did not decrease. A decrease was only apparent when treated with the λ-phosphatase (Fig 6C). Two-dimensional gel electrophoresis coupled with immunolabelling revealed that in the absence of stress, ZipD:3xHA resolved into a heterogeneous population of differentially charged species (Fig 6D). In response to calcium stress, ZipD:3xHA showed less electronegative species (more acidic isoforms), indicating protein dephosphorylation. To determine if calcineurin was important for ZipD:3xHA dephosphosphorylation, mycelia were exposed to either cyclosporin, or cyclosporin plus calcium. In response to cyclosporin plus calcium, ZipD:3xHA showed a less electronegative species (Fig 6D), consistent with calcineurin-independent protein dephosphorylation. These acidic isoforms disappeared when the protein extract was treated with λ-phosphatase, confirming they were phosphorylated isoforms (Fig 6D). Upon calcium stress, there is no physical interaction between CalA (the catalytic subunit of calcineurin) and ZipD:3xHA (S4 Fig).

To determine the effects of ZipD phosphorylations on *A. fumigatus* stress tolerance, we mutated ZipD on S54, S58, and S60, all three together to alanine (ZipD S/A) or to phosphomimetic aspartate (ZipD S/D; Fig 6E–6G). The ZipD:GFP, ZipD S/A, and ZipD S/D translocate to the nucleus in 9.3, 89.7, and 9.7%, respectively, of all counted hyphal germlings when they are incubated in absence of calcium (Fig 6E). In high concentrations of calcium, ZipD:GFP, ZipD S/A, and ZipD S/D translocate to the nucleus in 100, 91, and 9.3% (Fig 6E). Furthermore, the ZipD S/A strain was more sensitive to calcium and caspofungin stress, while the ZipD S/D strain presented a growth phenotype similar to the wild-type strain (Fig 6F and 6G). Together, these results confirm that ZipD is dephosphorylated upon calcium exposure, but not directly by calcineurin. However, its phosphorylation state determines nuclear translocation upon calcium stress, which is vital for resistance to calcium and caspofungin.

## Screening phosphatases involved in ZipD dephosphorylation

Assuming that a phosphatase null mutant involved in ZipD dephosphorylation upon calcium exposure would be more sensitive to $CaCl_2$, osmotic stress, and/or caspofungin, a library of 24 *A. fumigatus* non-essential phosphatase catalytic subunit null mutants [46] was screened for calcium sensitivity. Three phosphatase mutants (ΔsitA, ΔpptA, and ΔptcA) were more sensitive to $CaCl_2$ than the wild-type strain (S5 Fig). SitA (Afu6g11470) is the homologue of *S. cerevisiae* Sit4p, a cytoplasmic and nuclear phosphatase that modulates functions mediated by Pkc1p including cell wall and actin cytoskeleton organization [47]; PptA (Afu5g06700) is the homologue of *S. cerevisiae* Ppt1p that regulates Hsp90 chaperone by affecting its ATPase and cochaperone binding activities [48] while PtcA (Afu1g15800) is the homologue of *S. cerevisiae* Ptc6p

a mitochondrial phosphatase involved in mitophagy [49]. Only Δ*sitA* was more sensitive to osmotic stress tolerance and caspofungin than the wild-type and Δ*sitA*::*sitA*⁺ strains (S5 Fig). Considering its phenotypes and its involvement in the *A. fumigatus* CWI [50], the influence of SitA on ZipD dephosphorylation was assessed. Two-dimensional gel electrophoresis coupled with immunolabelling showed identical patterns of ZipD:3xHA heterogeneous population of differentially charged species (in the control) and less electronegative species (in response to calcium stress) in both wild-type and Δ*sitA* strains (S5 Fig). These results indicate that SitA is either not, or multiple phosphatases are, involved in ZipD dephosphorylation upon calcium stress.

In another attempt to identify phosphatase(s) involved in ZipD dephosphorylation, we hypothesize that since the expression of multiple genes is deregulated by the absence of ZipD during calcium stress, it was possible that phosphatase mutants involved in this process would also display comparable gene deregulation. Accordingly, the wild-type, Δ*zipD*, 24 non-essential phosphatase null mutants, and 4 essential phosphatase conditional mutants, were evaluated (S6 Fig). As previously observed, the *tcsA* (Afu6g10240) gene showed increased mRNA accumulation (about 2-fold) in the Δ*zipD* mutant when compared to the wild-type strain (Figs 4 and S2 and S6). Among the phosphatase null mutants Δ*sitA* (Afu6g11470), Δ*pypA* (Afu4g04710), Δ*dspD* (Afu2g02760), Δ*nemA* (Afu1g09460), Δ*ptcH* (Afu4g00720), Δ*ppsA* (Afu5g11690), Δ*prsA* (Afu1g04790), and Δ*pptA* (Afu5g06700) all showed higher *tcsA* mRNA accumulation than wild-type strain and Δ*zipD* mutant, suggesting they could be potentially involved in ZipD dephosphorylation (S6 Fig). Considering that SitA is not involved in ZipD dephosphorylation (S5 Fig), PypA and PpsA are tyrosine phosphatases, NemA is involved in phospholipid biosynthesis [38], the most likely candidates for ZipD dephosphorylation are the putative phosphatases DspD (a MAP kinase dual-specifity phosphatase), PtcH (Hog1 phosphatase), PrsA (involved in the general stress response), and PptA (that regulates Hsp90) [38]. Interestingly, the Δ*calA* (calcineurin catalytic subunit) had no impact on *tcsA* expression (S6 Fig).

Four of the phosphatase catalytic subunit genes [*glcA* (Afu1g04950), *pphB* (Afu6g10830), *fcpA* (Afu3g11410) and *dspA* (Afu1g13040)] are essential and conditional mutants were constructed by replacing the endogenous promoters with the *niiA* promoter (from the *A. fumigatus* nitrite reductase gene) [46]. The *niiA* promoter is induced by sodium nitrate and repressed by ammonium tartrate. The phosphatase conditional mutants were grown in sodium nitrate, then transferred to ammonium tartrate to repress phosphatase expression before CaCl₂ exposure (S6A Fig). None of these phosphatase mutants, when repressed and exposed to CaCl₂, exhibited higher *tcsA* mRNA accumulation than the wild-type and Δ*zipD* strains (S6A Fig), suggesting they are not involved in ZipD dephosphorylation. Taken together our results strongly suggest that ZipD dephosphorylation could be performed by more than one phosphatase.

## The Δ*zipD* mutant has highly attenuated virulence in immunodeficient mice

In the neutropenic BALB/c murine model of invasive pulmonary aspergillosis, wild-type and *zipD*::*zipD*⁺ infection resulted in 90 and 100% mortality at 5 and 7 days post-infection, respectively (statistically identical according to the Log-rank (Mantel-Cox) and Gehan-Brestow-Wilcoxon tests, *p* values >0.05; Fig 7A). The Δ*zipD* mutant caused 0% mortality at 15 days post-infection, which was not statistical different to the PBS (Phosphate Buffer Saline) control according to the Mantel-Cox and Gehan-Brestow-Wilcoxon tests (*p* values < 0.05; Fig 7A). Fungal burden was measured by qPCR, showing that the Δ*zipD* strain did not grow within the

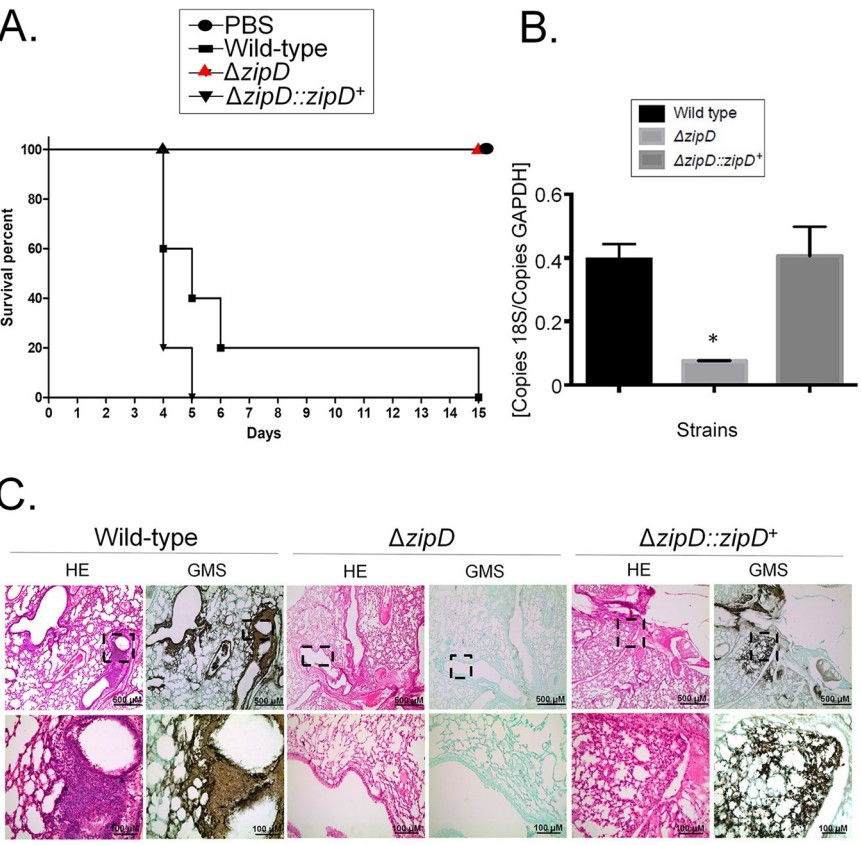

**Fig 7. Δ*zipD* has highly attenuated virulence in neutropenic mice.** (A) Comparative analysis of wild-type, Δ*zipD*, and Δ*zipD*::*zipD*⁺ strains in a neutropenic murine model of pulmonary aspergillosis. Mice in groups of 10 per strain were infected intranasally with a 20 μl suspension of conidia at a dose of 10⁵. PBS = phosphate Buffer Saline; *, *p*<0.001 comparison between the wild-type and Δ*zipD* mutant, for both Log−rank, Mantel−Cox and Gehan−Breslow −Wilcoxon tests. (B) Fungal burden was determined 48 hours post-infection by real-time qPCR based on 18S rRNA gene of *A. fumigatus* and an intronic region of the mouse GAPDH gene. Fungal and mouse DNA quantities were obtained from the Ct values from an appropriate standard curve. Fungal burden was determined through the ratio between ng of fungal DNA and mg of mouse DNA. The results are the means (± standard deviation) of five lungs for each treatment. Statistical analysis was performed by using *t*-test (*, *p* < 0.01). (C) Histopathology of mice infected with *A. fumigatus* wild-type, Δ*zipD* and Δ*zipD*::*zipD*⁺ mutant strains. GMS (Grocott's Methenamine Silver) and HE (Hematoxylin and Eosin) staining of lung sections of representative of infections. The hatched area from the top row is amplified in the lower row. Bars, 100 and 500 μm.

lungs as well as the wild-type and the complemented Δ*zipD*::*zipD*⁺ strains (Fig 7B). Histopathological examination revealed that at 72 h post-infection the lungs of mice infiltrated with either PBS or the Δ*zipD* strain showed no sign of inflammation or fungal burden. In contrast, mice infected with the wild-type or the Δ*zipD*::*zipD*⁺ strains contained multiple foci of invasive hyphal growth, which penetrated the pulmonary epithelium in major airways, while pockets of branched invading hypha originated from the alveoli (Fig 7C). These data strongly indicate that when compared to the wild-type strain the lack of ZipD in *A. fumigatus* caused a significant virulence reduction in immunodeficient mice.

## The Δ*zipD* mutant stimulates enhanced macrophage-mediated killing and proinflamatory cytokines

The Δ*zipD* mutation altered cell wall composition, which may influence the immune response and virulence. Macrophages contribute to innate immunity, fungal clearance and the

generation of a pro-inflammatory response during *A. fumigatus* infection [51]. Thus, the capacity of bone marrow-derived macrophages (BMDMs) to phagocyse and kill the wild-type, Δ*zipD* and Δ*zipD*::*zipD*⁺ conidia was assessed. The *zipD* deletion mutant conidia were more efficiently killed by BMDMs compared to the wild-type and Δ*zipD*::*zipD*⁺ conidia (Fig 8A). These results indicate that Δ*zipD* was more susceptible to macrophage killing.

The macrophage-mediated inflammatory response of BMDMs obtained from C57BL/6 mice exposed to the wild-type, Δ*zipD* and Δ*zipD*::*zipD*⁺ conidia was assessed (Fig 8B). Compared to wild-type and Δ*zipD*::*zipD*⁺ strains, the Δ*zipD* mutant induced the increased production of pro-inflammatory cytokines, such as IL-12p40, IL-6, IL-1β and TNF-α. Collectively, this shows that ZipD mediated regulation of the fungal cell wall may be important for evading the induction of the pro-inflammatory responses, and in turn promoting virulence.

### The Δ*zipD* mutant has highly attenuated virulence in immunocompetent mice associated with enhanced immune activation

We also decided to investigate the importance of ZipD to elicit immune responses by using an immunocompetent mouse model. The influx of leukocytes into the lungs of *A. fumigatus*-infected immunocompetent C57BL/6 mice was assessed. After 72 h infection, lung infiltrating leukocytes were obtained and analyzed for the expression of surface and intracellular molecules by flow cytometry. A higher number of neutrophils (CD11b⁺Ly6G⁺), macrophages and activated macrophages (CD11b⁺F4/80⁺MHC-II⁺ and CD11b⁺F4/80⁺CD86⁺) were found in the lungs of mice infected with the Δ*zipD* mutant when compared with wild-type infection (Fig 9A and 9B). The influx of T cell subpopulations into the lungs of infected mice also showed a higher number of both total (CD4⁺) and effector/activated (CD4⁺CD25⁺) CD4⁺T cells in Δ*zipD* infected mice, compared with wild-type and Δ*zipD*::*zipD*⁺ infections (Fig 9C). Similar observations were seen with the total CD8⁺ T cells, but no differences were found for activated CD8 T cells (CD8⁺CD69⁺) (Fig 9D).

The BMDMs studies showed that the Δ*zipD* mutant stimulated the production of higher levels of pro-inflammatory cytokines. Therefore, Th subpopulations were assessed in the lungs of infected immunocompetent C57BL/6 mice. Increased numbers of CD4⁺IFN-γ⁺and CD4⁺IL-17⁺ T cells, which define the Th1 and Th17 subsets respectively, were found in Δ*zipD* infected mice, in comparison to wild-type and Δ*zipD*::*zipD*⁺ infections. No differences were observed for the Th2 (CD4⁺ IL-4⁺ cells) and Tc (CD8⁺) subpopulations (Fig 9F). Lung cytokine concentrations were also measured by ELISA. After 72 h infection, the lungs of Δ*zipD* infected mice showed augmented levels of inflammatory cytokines, including IFN-γ, IL-12 and IL-6, when compared with wild-type and Δ*zipD*::*zipD*⁺ infections. The Th17 cytokine, IL-17, exhibited a similar but non-significant trend (Fig 10A).

Histological examination revealed similar observations to those obtained with the immunodeficient mice. The lungs of Δ*zipD* infected mice showed reduced fungal burden, while wild-type or Δ*zipD*::*zipD*⁺ infections presented multiple foci of invasive hyphal growth. In agreement with flow cytometry data, histological examinations revealed that the Δ*zipD* infected mice presented a strong cellular influx to the lungs (Fig 10B). Together, the data obtained from the immunocompetent mice model strongly indicates that the absence of ZipD caused a significantly enhanced activation of innate and adaptive immunity, which may contribute to a reduction in *A. fumigatus* infection.

## Discussion

Calcium signaling regulates fungal growth, metabolism, morphogenesis, cell wall and osmotic stress tolerance, and virulence [9, 52]. Additionally, the calcium-calcineurin pathway, through

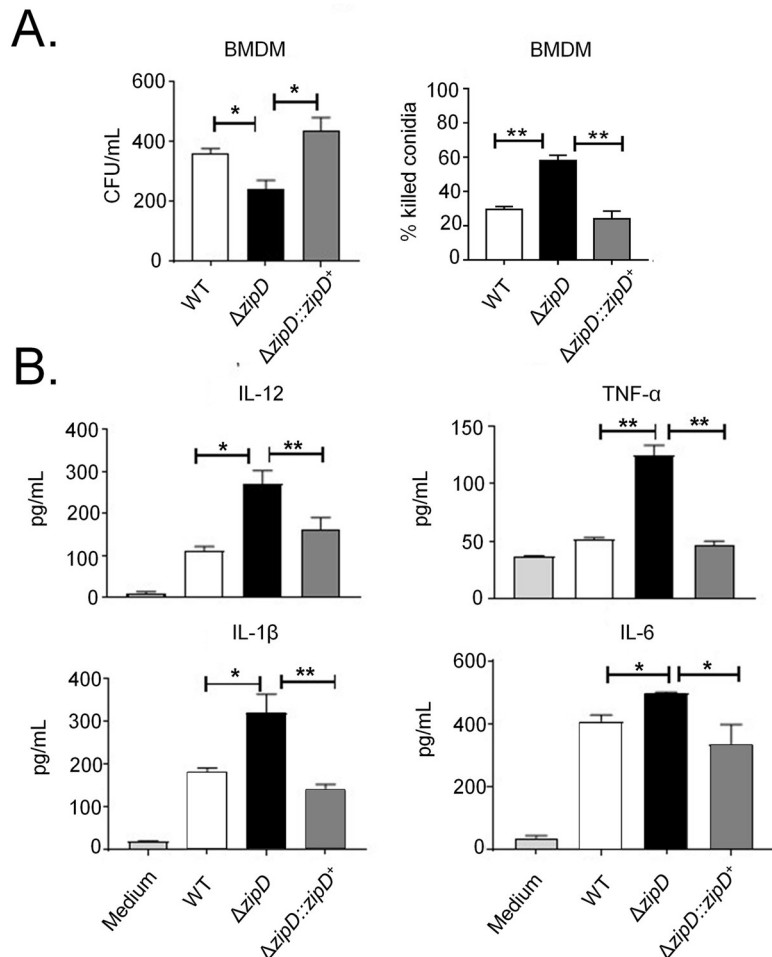

**Fig 8. ΔzipD is more sensitive to macrophage killing and elicits an increased immune response.** (A) Fungal killing capacity of mouse bone marrow-derived macrophages (BMDMs) assessed as colony forming units (CFU) of *A. fumigatus* after 4 h of infection (left panel). The percentage killing was calculated from CFU remaining compared with control samples without macrophages (right panel). (B) IL-12p40, IL-1β, IL-6, and TNF−α secretion by BMDMs. BMDMs from C57BL/6 mice were infected with *A. fumigatus* conidia and cocultivated for 4 h to allow conidia adhesion and ingestion. The monolayers were washed with PBS to remove any non-ingested or non-adhered conidia and the samples were cultured for an addition period of 18 h to permit the cytokine production by adherent cells. The supernatant was collected to measure the cytokine levels by ELISA. Data are the mean ± SD (Standard Deviation) of quintuplicate samples from one experiment representative of three independent determinations (*$p < 0.05$ and **$p < 0.01$).

its influence on the cell wall, promotes caspofungin resistance and the CPE, highlighting the clinical importance of the calcium signaling network to the treatment of life-threatening *A. fumigatus* infection. For years, in *A. fumigatus* CrzA was recognized as the single calcium/calcineurin-dependent TF. Now, eight additional TFs, including ZipD, have been discovered to influence calcium signaling and the CPE [32]. The genes Afu1g10550, Afu5g10620, Afu4g07090 are without any known functional annotation, whereas Afu7g03910 (*nsdC*) is the *A. nidulans* NsdC homologue required for sexual development [53]. Finally, Afu6g12522 (*skn7*) contributes to growth in the presence of hydrogen peroxide and tert-butyl hydroperoxide induced oxidative stress, but it is not involved in virulence [54]. Interestingly, two null mutants that displayed deficient growth phenotypes were partially rescued by the addition of calcium. These included the novel Afu1g13190 gene and Afu3g08010 (*ace1*) the homologue of

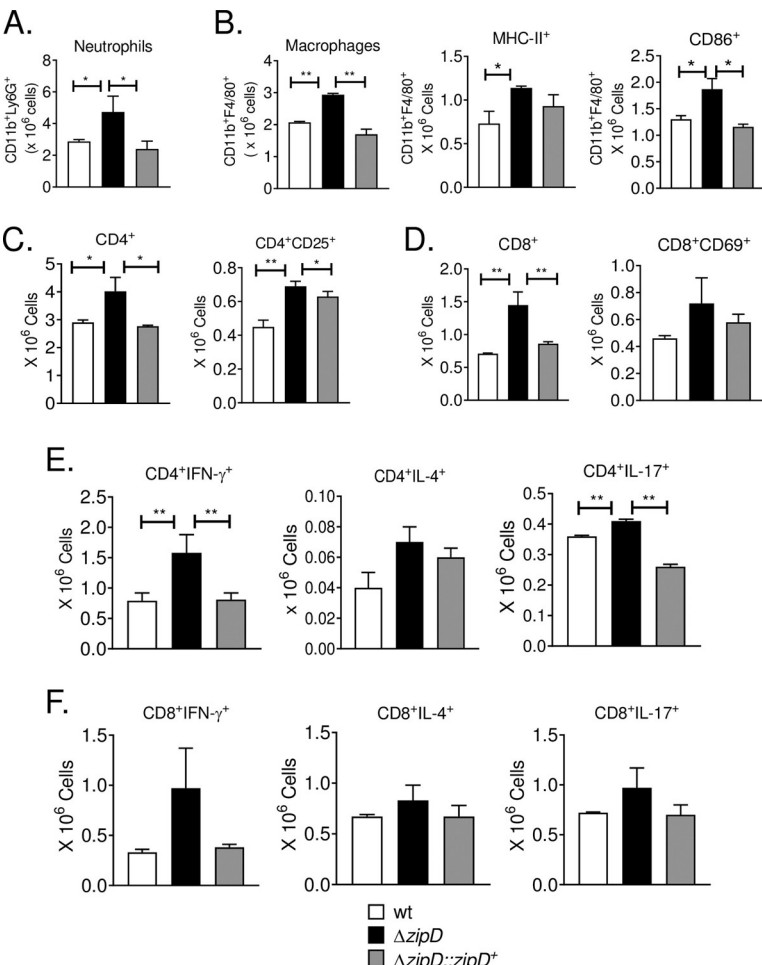

**Fig 9. Δ*zipD* exhibits increased pulmonary neutrophils, macrophages and activated T CD4 lymphocytes in immunocompetent mice.** C57BL/6 mice were infected with 5 x 10⁷ *A. fumigatus* conidia by *i.t.* (intra-tracheal) route and cell phenotypes determined after 72 h of infection. Lungs from groups of mice (n = 3–4) were excised and digested enzymatically. Cell suspensions were obtained and stained as described in Materials and Methods. The stained cells were analyzed immediately on a FACSCanto II equipment with gating of lymphocytes or granulocytes, as judged from FSC and SSC scatters. The number of total neutrophils CD11b$^+$F4/80$^-$Ly6G$^+$ (A) macrophages F4/80$^+$CD11b$^+$ (B) and activated CD4$^+$ T (C) and CD8$^+$ T (D) cells was assessed. The presence of IFN-γ$^+$, IL-4 and IL-17$^+$ in CD4$^+$ and CD8$^+$ T cells in the lung infiltrating leukocyte was also assessed. Lung cells were stimulated *in vitro* with PMA/ionomycin for 6 h and subjected to intracellular staining for cytokines. Lymphocytes were gated for CD4 (Fig E) or CD8 (Fig F) and then for IFN-γ, IL-4 and IL-17 expression. One hundred thousand cells were counted, and the data expressed as number of positive cells. Data are expressed as means ± standard error of the mean and are representative of two independent experiments (*$p < 0.05$ and **$p < 0.01$).

*A. nidulans* SltA involved in cation homoeostasis and detoxification [55]. It remains to be determined if these identified genes perform the same functions in the two *Aspergillus* species. These results expand significantly the number of TFs involved, and suggest complex mechanisms in regulating calcium metabolism.

ZipD dephosphorylation is shown to be essential for its nuclear translocation. Although, calcineurin is important for ZipD functionality and nuclear translocation upon calcium or caspofungin exposure [32], ZipD dephosphorylation is calcineurin-independent and ZipD does not physically interact with calcineurin. Therefore, to identify the phosphatase responsible for ZipD dephosphorylation upon calcium stimulation, the *A. fumigatus* library of non-essential

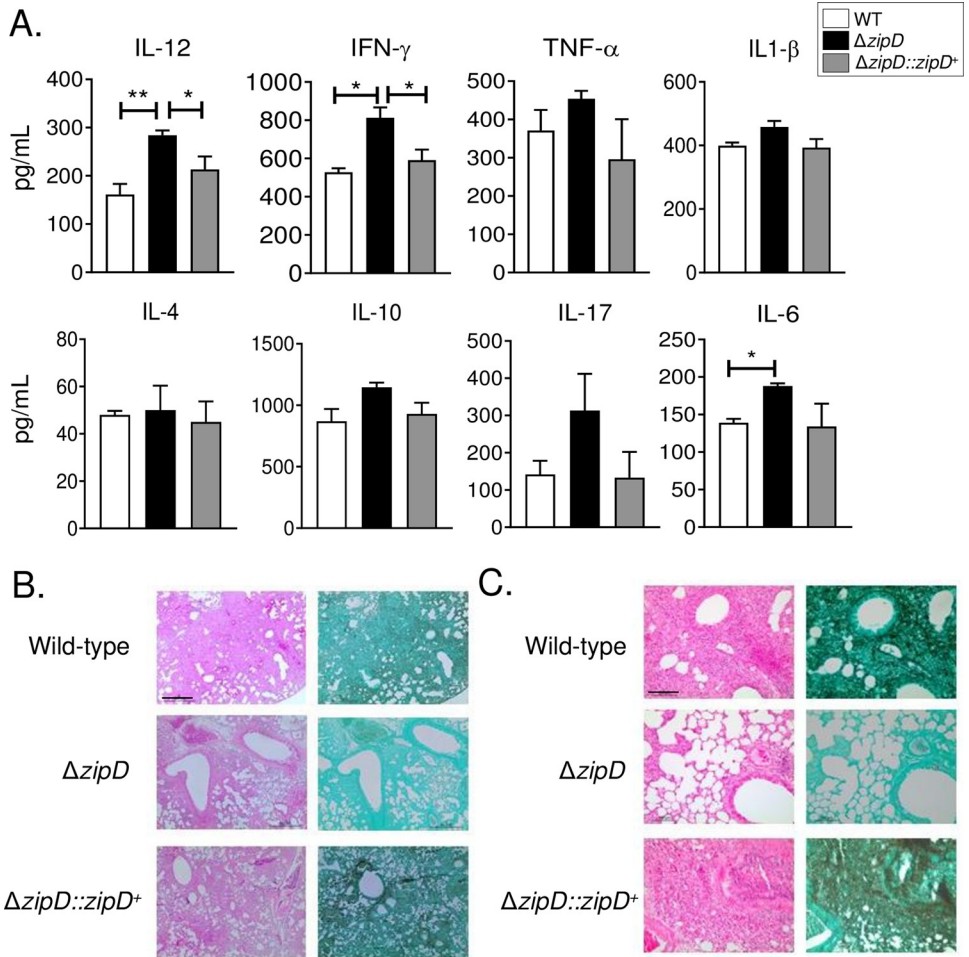

**Fig 10. Δ*zipD* elicits increased production of pro-inflammatory cytokines in immunocompetent mice.** (A) Cytokines quantitation by ELISA in lung homogenates from C57BL/6 mice after 72 h of infection with 5 x 10$^7$ *A. fumigatus* conidia by *i.t.*(intra-tracheal) route. Bars show mean ± SD (Standard Deviation) from at least four mice per group and are representative of two independent experiments ($^*p < 0.05$ and $^{**}p < 0.01$). (B-C) Histopathology of mice infected with *A. fumigatus* wild-type, Δ*zipD* and Δ*zipD::zipD*$^+$ mutant strains. GMS (Grocott's Methenamine Silver) and HE (Hematoxylin and Eosin) staining of lung sections of representative infections. Bars, 500 μm (B; GMS, right panel; HE, left panel) and 100 μm (C; GMS, right panel; HE, left panel).

null phosphatase mutants was screened for sensitivity to calcium, osmotic stress, and decreased CPE. This identified SitA as a putative ZipD dephosphorylating phosphatase, since the Δ*sitA* mutant was more sensitive to all these three phenotypic conditions and has previously been shown to be involved in the CWI pathway [50]. However, further investigation revealed in the absence of SitA, ZipD could still be dephosphorylated. Therefore, ZipD dephosphorylation appears to be performed by more than one phosphatase or an essential phosphatase [46].

To evaluate ZipD functionality the expression of ZipD-regulated *tcsA* expression was assessed in multiple null non-essential and conditional essential phosphatases. The *tcsA* gene has increased mRNA accumulation in the presence of calcium in Δ*zipD* than in the wild-type. The four essential phosphatases appear not to be involved. However, eight null non-essentail phosphatase mutants showed higher *tcsA* mRNA levels than the Δ*zipD*, suggesting these phosphatases could be involved in ZipD dephosphorylation and its subsequent activation. Four of

these phosphatases, DspD, PtcH, PrsA, and PptA, which are important for MAP kinase modulation, general stress responses and Hsp90 regulation, are the most likely candidates for ZipD dephosphorylation [38]. This approach also revealed calcineurin did not regulate *tscA* expression, differentiating aspects of calcineurin signaling and ZipD function. This suggests that while ZipD nuclear translocation is calcineurin-dependent multiple other phosphatases could be involved in ZipD dephosphorylation and activation. Since there are three different serine phosphorylation sites in the ZipD, it is possible there is more than one phosphatase responsible for ZipD dephosphorylation upon calcium or caspofungin exposure. It remains to be determined which protein kinases and phosphatases are phosphorylating/dephosphorylating ZipD upon calcium and caspofungin stimulation.

Transcriptional profiling defined the overlapping roles of ZipD and CrzA regulating calcium homeostasis, in addition to ZipD-specific functions in cell wall organization and osmosensing. Although these initial assays used high $CaCl_2$ concentrations that could potentially induce osmotic stress, multiple additional growth assays, in the absence of $CaCl_2$, validate that ZipD is involved in osmotic stress tolerance. Fungal genetics confirmed ZipD and CrzA additively interact in parallel pathways, in response to high calcium concentrations. Additionally, both ZipD and CrzA are translocated to the nucleus upon osmotic stress, modulating the expression of genes involved in the osmotic stress (HOG) pathway. CrzA directly binds to and activates several members of the HOG pathway [30]. This suggests that both ZipD and CrzA influence the expression of genes in the signaling pathways important for their own activation. However, it is ZipD that plays the major role in the response to cell wall damage and osmotic stress, through its modulation of cell wall structure. This suggest a complex interaction network exists between the calcineurin, CrzA, ZipD and HOG pathways, which are essential for a proper response to cell wall damage and osmotic stress in A. *fumigatus*.

The mammalian host environment has approximately 2 mM calcium [56]. This high level of calcium affects fungal metabolism and requires the human pathogenic fungus A. *fumigatus* to tightly regulate calcium homeostasis, via the calcineurin-CrzA pathway. This mechanism allows A. *fumigatus* to keep cytosolic free calcium at very low (50–100 nM) levels [57]. CrzA is essential for virulence in an experimental neutropenic inhalational murine model of invasive pulmonary aspergillosis [26, 28]. Similarly, the absence of ZipD causes highly attenuated virulence, which could reflect an inability to adapt to high calcium concentrations in the host. However, the absence of ZipD also increases sensitivity to cell wall damaging agents, osmotic stress and caspofungin. ZipD has opposing affects on major cell wall polysachharides, reducing β-1,3-glucans and increasing chitin content in the cell wall. Nonetheless, the absence of ZipD increases cell wall thickness, while reducing both β-1,3-glucan and chitin exposure. Therefore, ZipD influences cell wall structure and resistance to cell wall-related stresses, which may promote the survival of A. *fumigatus* within the host and contribute to caspofungin resistance.

The A. *fumigatus* osmotic stress (HOG) pathway is composed of two signaling modules: (i) the two-component system (TCS)-like phosphorelay module and (ii) the MAP kinase module [58]. ZipD was identified as another transcription factor, which in addition to AtfA, B, C, and D [41, 59], is involved in the osmotic stress response. Both ZipD and CrzA are translocated to the nucleus upon osmotic stress and modulate the expression of genes encoding for proteins in the TCS and the HOG pathway. This suggests that both transcription factors influence the expression of genes in the signaling modules important for their own activation. A ChIP-seq (Chromatin Immunoprecipitation DNA sequencing) analysis previously showed that CrzA directly influences several members of the TCS and the HOG pathway [16]. Interestingly, Δ*crzA* was not more sensitive to osmotic stress, caspofungin, or cell wall damaging agents, which could reflect the redundancy between the activated transcription factors, including ZipD which was shown to play a major role in the osmotic stress response.

Furthermore, the defects in cell wall structure caused by the absence of ZipD increased macrophage phagocytosis and killing, while stimulating an enhanced pro-inflammatory response and the influx of leukocytes to the site of infection. In response to infection in the absence of ZipD, the enhanced production of lung-infiltrating neutrophils associated with Th17 immunity, as well as the activated macrophages associated with Th1 immunity, within the bronchoalveolar space would contribute to improved fungal clearance and reduced virulence. The high number of neutrophils and Th17 cells suggests that IL-17A is produced by T cells, which promote the accumulation and enhanced activity of neutrophils in the bronchoalveolar space to fight against fungal infections [60–62]. This suggests that ZipD is important for the evasion of immune activation and virulence, potentially through its regulation of fungal cell wall structure. Together, this host immunological data suggests that the alterations in *A. fumigatus* cell wall composition caused by the Δ*zipD* mutation influences immune activation and pathogenesis, as cell wall polysaccharides combinations can activate different inflammatory responses [63].

There is limited knowledge of *A. fumigatus* TFs regulating cell wall biosynthesis and/or remodelling. *A. fumigatus* TFs such as RlmA, CrzA, and ZipD have been characterized in more detail as regulating several genes involved in the cell wall metabolism [9, 14–15, 26–28, 30, 32, 64]. These modifications in the cell wall are maybe vital during pathogenesis as they could affect the interaction and recognition by the immune system from a non-protective and weak, to an over-inflamatory and damaging response. An interesting comparison to this study is the *Cryptococcus neoformans* Rim101 transcription factor, an important regulator of the adaptive cell surface changes that occur during infection [65]. Ost *et al.* [66] demonstrated that the *rim101Δ* mutant has increased chito-oligomer exposure on its cell surface in response to host-like conditions, leading to increased recognition by macrophages *in vitro* and increased inflammation *in vivo*. Similar to Δ*zipD*, *rim101Δ* cell wall increases macrophage detection in a contact-dependent manner and elicits an increase in Th1 and Th17 cytokine responses. These results suggest that both TFs are important to regulate their cell walls composition and organization and prevent the exposure of immune stimulatory molecules within the host.

This study emphasizes the complexity of the calcium/calcineurin network, revealing additional layers of regulation, which control the fungal response to multiple stresses requiring the cell wall remodeling. These functions in turn contribute to virulence and antifungal resistance. This reinforces the idea that components of the fungal cell wall and their regulatory mechanisms are potential drug targets for the development of novel combinational therapies for aspergillosis.

## Methods

### Ethics statement

Animal experiments were performed in strict accordance with the Brazilian Federal Law 11,794 establishing procedures for the scientific use of animals in strict accordance with the principles outlined by the Brazilian College of Animal Experimentation (CONCEA), and the State Law establishing the Animal Protection Code of the State of São Paulo. All protocols adopted in this study were approved by the local ethics committee for animal experiments from the University of São Paulo, Campus of Ribeirao Preto (Permit Number: 08.1.1277.53.6; Studies on the interaction of *Aspergillus fumigatus* with animals). All efforts were made to minimize suffering. Animals were clinically monitored at least twice daily and humanely sacrificed if moribund (defined by lethargy, dyspnea, hypothermia and weight loss). All stressed animals were sacrificed by cervical dislocation after intraperitoneal (i.p.) injection of ketamine and xylazine.

## Strains and media

The *A. fumigatus* strains used in this study are CEA17, A1160, Af293, $\Delta crzA^{CEA17}$, $\Delta crzA$-$^{CEA17}$::$crzA^+$, $\Delta zipD^{CEA17}$, and $\Delta zipD^{CEA17}$::$zipD^+$, ZipD:GFP, ZipD:GFP S/A, and ZipD:GFP S/D. The phosphatase mutants are described in the S10 Table. Strains were grown at 37˚C in either complete medium [CM: 2% (w/v) glucose, 0.5% (w/v) yeast extract, trace elements] or minimal medium [MM: 1% (w/v) glucose, original high nitrate salts, trace elements, pH 6.5]. Solid CM and MM were the same as described above except that 1.7% (w/v) or 2% (w/v) agar was added. Where necessary, uridine and uracil (1.2 g/L) were added. Trace elements, vitamins, and nitrate salts compositions were as described previously [67].

## Generation, validation, and screening of transcription factor knockout mutants

The transcription factor knockout mutant collection was generated in the *A. fumigatus* strain A1160 (a derivative of CEA17, ΔKu80 pyrG$^+$) according to Furukawa *et al.* [68]. Gene knockout cassettes were generated using a fusion PCR approach. Briefly primers P1 and P2 were used to amplify around 1 kb of the 5' flank, while P3 and P4 were used to amplify the 3' flank. Primers hph_F and hph_R were used to amplify a 2.8 kb hygromycin B phosphotransferase cassette from pAN7.1. PCR products were purified by solid phase extraction (Qiagen QIA-quick PCR purification kit). Fusion of the three products was facilitated by the presence of common linker sequences on primers P2 and hph_F, P3 and hph_R and the use of the nested primers P5 and P6. The sequences of all primers used are given in S11 Table. Validation of homologous recombination and single integration of the deletion cassette was performed by PCR. PhusionFlash High-Fidelity Master mix (ThermoFisher) was used for all reactions. Primers P1 with hph-chk 5'-Rv and hph-chk 3'-Fw with P4 were used to amplify a region of about 1kb from within the deletion cassette to the flanking region outside of the deletion cassette. Furthermore, PCR was performed with P1 and P4 as primers to check the purity. The mutants were grown in liquid and solid MM at 37˚C for 5 days. The most sensitive and resistant mutants were identified, retested, and purified for further characterization.

## Construction of the *A. fumigatus* mutants

To generate the ZipD:3xHA::pyrG fusion fragment, a 1.6-Kb portion of DNA consisting of the zipD ORF and 5' UTR region, along with a 1-Kb segment of DNA consisting of the 3' UTR flanking region were amplified with primers zipD pRS426 5fw(3xHA)/zipD orf LINKER 3HA rv and zipD 3utr pyrG 3fw/zipD pRS426 3rv, respectively, from CEA17 gDNA. The 2.7-kb 3xHA::pyrG fusion was amplified with primers OZG916/OZG964 from the pOB430 plasmid. The cassette was generated by transforming each fragment along with the plasmid pRS426 cut with BamHI/EcoRI into the *S. cerevisiae* strain. This cassette was then transformed into the CEA17 strain and verification of ZipD tagging was confirmed via PCR reaction S3 Fig). The ZipD phosphomutants were constructed by replacing the serines at 54, 58 and 60 residues by Alanine (A) or Glutamic Acid (D). Mutations were inserted by using the oligonucleotides: 54 mut zip S-A Fw, 58 mut zip S-A Fw, 60 mut zip S-A Fw and mut 54-58-60 zip S-A Fw for serine to alanine substitution; and 54 mut zip S-A Fw, 58 mut zip S-D Fw, 60 mut zip S-D Fw and mut 54-58-60 zip S-D Fw for serine to aspartic acid substitution. Two fragments were obtained: about 1-kb fragment consisting of 5' UTR region and zipD ORF and a ~4.6kb fragment of the ZipD ORF + GFP-trpC-prtA + 3' UTR flanking region. This cassette was generated by transforming these two fragments along with the plasmid pRS426 into the *S. cerevisiae* strain, following by transformation into electrocompetent *E. coli* DH10B strain. The plasmid

DNA of the transforming bacteria was extracted and sequenced to verify the mutations. The plasmid was digested with *Stu*I and *Dra*I enzymes to release the cassette, which was subsequently transformed into the CEA17$^{\Delta ku80}$ strain. The primers used above are described in S11 Table.

## Immunoblot analysis

To assess the phosphorylation status of SakA and MpkA, fresh harvested conidia ($1x10^7$) of the wild-type and mutant strains were inoculated in 50 ml liquid MM at 37˚C for 16 h (180 rpm). After incubation, the cultures were transferred to MM+1.2 M Sorbitol or CR (300 μg/ml) for 10 and 30 minutes. Control was left untreated. Mycelia were ground in liquid nitrogen with pestle and mortar. For protein extraction, 0.5 ml lysis buffer containing 10% (v/v) glycerol, 50 mM Tris–HCl pH 7.5, 1% (v/v) Triton X-100, 150 mM NaCl, 0.1% (w/v) SDS, 5 mM EDTA, 50 mM sodium fluoride, 5 mM sodium pyrophosphate, 50 mM β-glycerophosphate, 5 mM sodium orthovanadate, 1 mM PMSF, and 1X Complete Mini protease inhibitor (Roche Applied Science) was added to the ground mycelium. Extracts were centrifuged at 20,000 g for 40 minutes at 4˚C. The supernatants were collected and the protein concentrations were determined using the Bradford assay (BioRad). Fifty μg of protein from each sample were resolved in a 12% (w/v) SDS–PAGE and transferred to polyvinylidene difluoride (PVDF) membranes (Merck Millipore). The total and phosphorylated fractions of the MAP kinase, MpkA, were examined using anti-p44/42 and anti-phospho p44/42 MAPK antibody (Cell Signaling Technologies) following the manufacturer's instructions using a 1:1000 dilution in TBST buffer (137 mM NaCl, 20 mM Tris, 0.1% Tween-20). To detect SakA and phosphorylated SakA proteins, a mouse polyclonal IgG antibody against Hog1 y-215 (Santa Cruz Biotechnology, Santa Cruz, CA, USA) and a rabbit polyclonal IgG antibody against dually phosphorylated p38 MAPK (Cell Signaling Technology, Beverly, MA, USA) were used, respectively. Primary antibody was detected using an HRP-conjugated secondary antibody raised in rabbit (Sigma). Chemoluminescent detection was achieved using an ECL Prime Western Blot detection kit (GE HealthCare). To detect these signals on blotted membranes, the ECL Prime Western Blotting Detection System (GE Healthcare, Little Chalfont, UK) and LAS1000 (FUJIFILM, Tokyo, Japan) were used. The images generated were subjected to densitometric analysis using ImageJ software (https://imagej.nih.gov/ij/index.html).

## Immunoprecipitation (IP) with anti-HA magnetic beads and *in vitro* phosphatase assay

To perform IP assays, C-terminal HA-tagged ZipD strain was generated in the CEA17$^{\Delta ku80}$ background. IP experiment was performed in the same way as the previously described [69]. Briefly, mycelia were frozen with liquid nitrogen and ground, and 500 mg was resuspended in 1 ml of B250 buffer [70]. Samples were centrifuged at maximum speed for 10 min at 4˚C. Supernatant was removed, and a Bradford assay (BioRad) was carried out to measure protein content. The same amount of protein for each sample was added to 20 μl of Dynabeads Protein A (Thermo Fisher Scientific) previously incubated with monoclonal anti-HA antibody (Sigma). The resin was washed three times with resuspension buffer prior to incubation. Cell extracts and resin were incubated with shaking at 4˚C for 2 h. After incubation, the resin was washed three times in resuspension buffer by placing the tube in a DynaMag magnet. To release the proteins from the resin, samples were incubated with sample buffer and boiled at 98˚C for 5 min. Proteins were transferred from a 10% SDS-PAGE gel onto a nitrocellulose membrane for a Western blot assay using a Trans-Blot turbo transfer system (Bio-Rad). HA-tagged ZipD was detected using a mouse monoclonal anti-HA antibody (Sigma) at 1:2,000

dilution and a goat anti-mouse IgG HRP antibody (Cell Signaling Technology) at 1:10,000 dilution. For bidimensional electrophoresis, the protein extraction and immunoprecipitation were performed as described above, except by the release of the proteins from the resin by the addition of Rehydration buffer (7M Urea, 2M Thiourea, 2% CHAPS, 0.35mg/sample DTT) (for Cyclosporin-treated samples, the elution was carried out by addition 0.1M Glycine pH3.0, followed by protein precipitation with four volumes of acetone at -20˚C overnight, and resuspension with Rehydration buffer). The isoleletric focalization (IEF) was performed using Ettan IPGphor 3 System (GE Healthcare), with 7 cm DryStrip (pH 4–7) immobiline strips. The IEF was carried out following the default steps suggested in GE Healthcare 2D instructions (300V for over 200Vh; ramping 1000V for over 300Vh and 5000V for 450Vh; holding at 5000V for 3000Vh). The second dimension was performed by incubation of the strips with 1% DTT in equilibration buffer (75 mM Tris-HCl pH 8.8; 6 M urea; 29% glycerol and 2% SDS) for 15 min followed by 15 min in equilibration buffer with 2.5% iodoacetamide. The strips were washed in water and transferred to the top of 12% SDS-Polyacrylamide gel, covered with 0.5% agarose solution and submitted to electrophoresis. After running, the proteins from the gels were transferred to a nitrocellulose membrane for a western blot assay, following the steps described above for detection of HA-tagged ZipD. Phosphorylated ZipD:3xHA was detected by immunoblotting with Anti-Phosphoserine/threonine/tyrosine antibody (Abcam) at a 1:2000 dilution.

The lambda phosphatase assay was performed as previously described [71]. The ZipD-3xHA immunoprecipitates bound to magnetic beads were incubated with PMP buffer (50 mM HEPES pH 7.5, 100 mM NaCl, 2 mM DTT, 0.01% Brij 35, and 1 mM MnCl2) and 400 units of lambda protein phosphatase (New England BioLabs) (without phosphatase inhibitor cocktail) for 1 hour at 30˚C. The ZipD-3xHA magnetic beads were resuspended in sample buffer, boiled for 5 min, briefly centrifuged, and the supernatant was resolved by SDS-PAGE and transferred to nitrocellulose membranes (Bio-Rad). Membranes were assayed by western blot employing mouse monoclonal anti-HA antibodies (Sigma), followed by anti-mouse antibody conjugated to HRP, and ECL western blotting detection reagent (GE Healthcare).

### In solution digestion and peptide desalting

Immunoprecipitate samples from Zip::HA control and CaCl$_2$-treated were re-suspended in 8M urea, 100mM Ambic and incubated under stirring for 30 min to solubilize the proteins. Proteins were reduced with 10 mM DTT (DL-Dithiothreitol–Sigma-Aldrich) for 30 min at 30˚C, alkylated with 40 mM iodoacetamide (Sigma-Aldrich) for 30 min in the dark and digested with trypsin (Promega) in the ratio 1:50 (μg trypsin/μg protein) in 50 mM ammonium bicarbonate solution at 30˚C overnight. The reaction was stopped with 1% formic acid (less than pH 3) and then the sample was desalted with C18 columns (StageTips) [72].

### Nano LC-MS/MS analysis

Peptide samples were resuspended in 0.1% formic acid (FA) before analysis using a nano-flow EASY-nLC 1200 system (Thermo Scientific) coupled to Orbitrap Fusion Tribrid mass spectrometer (Thermo Scientific). The peptides were loaded on an Acclaim PepMap C18 (Thermo Germany) trap column (2 cm x 100 μm inner diameter; 5 μm) and separated onto an Acclaim PepMap C18 (15 cm x 75 μm inner diameter; 3 μm) column and separated with a gradient from 100% mobile phase A (0.1% FA) to 28% phase B (0.1% FA, 80% ACN) during 70 min, 28%-40% in 10 min, 40%-95% in 2 min and 12 min at 95% at a constant flow rate of 300 nL/min. The mass spectrometer was operated in positive ion mode with data-dependent acquisition. The full scan was acquired in the Orbitrap at a resolution of 120,000 FWHM in the 375–

1600 m/z mass range with max injection time of 50ms and AGC target of 5e5. Peptide ions were selected using the quadrupole with an isolation window of 1.2 and fragmented with HCD MS/MS using a normalized collision energy of 35 and detected in the ion trap. Data dependent acquisition with a cycle time of 3 seconds was used to select the precursor ions for fragmentation. Dynamic exclusion was activated with 12 sec as exclusion duration and 20ppm as mass tolerance. All raw data were accessed in Xcalibur software (Thermo Scientific).

## Database searches and bioinformatics analyses

Raw peptide data were processed using MaxQuant software version 1.5.2.8 and the embedded database search engine Andromeda [73]. The MS/MS spectra were searched against Uniprot Aspergillus fumigatus Protein Database (downloaded October, 2017; 9648 entries), with the addition of common contaminants, with an MS accuracy of 4.5 ppm and 0.5 Da for MS/MS. Cysteine carbamidomethylation (57.021 Da) was set as the fixed modification, and two missed cleavages for trypsin. Methionine oxidation (15.994 Da), protein N-terminal acetylation (42.010 Da) and phosphorylation S/T/Y (+79.96 Da) were set as variable modifications. Proteins and peptides were accepted at FDR less than 1%. Proteins with at least two peptides and two ratio counts were accepted for further validation. Label-free quantification was performed using the MaxQuant software with the "match between run" and iBAQ features activated. The MS intensity of phosphopeptides was compared between the different conditions.

## Staining for dectin-1 and chitin

Cell wall surface polysaccharide straining was performed as described previously [74, 75]. Briefly, strains were grown from $2.5 \times 10^3$ spores in 200 μl of MM for 16 h at 37°C before the culture medium was removed and germlings were UV-irradiated (600,000 μJ). Hyphal germlings were subsequently washed with PBS before 200 μl of a blocking solution [2% (w/v) goat serum, 1% (w/v) BSA, 0.1% (v/v) Triton X-100, 0.05% (v/v) Tween 20, 0.05% (v/v) sodium azide and 0.01M PBS] were added and samples were incubated for 30 min at room temperature (RT).

For dectin staining, 0.2 μg/ml of Fc-h-dectin-hFc were added to the UV-radiated germlings and incubated for 1 h at RT, followed by the addition of 1:1000 DyLight 594-conjugated, goat anti-human IgG1 for 1 h at RT [58]. Germlings were washed with PBS and fluorescence was read at 587 nm excitation and 615 nm emission. For chitin staining, 200 μl of a PBS solution with 10 μg/ml of CFW were added to the UV-irradiated germlings, incubated for 5 min at RT, washed three times with PBS before fluorescence was read at 380 nm excitation and 450 nm emission. All experiments were performed using 12 repetitions and fluorescence was read in a microtiter plate reader SpectraMax i3 (Molecular Devices).

## Cell wall polysaccharides extraction and sugar quantification

Fungal cell wall polysaccharides were extracted from 10 mg dry-frozen biomass as described previously [76]. One mL of extracted samples were concentrated 10 x by liophilization and sugars subsequently analyzed by HPLC using a YoungLin YL9100 series system (YoungLin, Anyang, Korea) equipped with a YL9170 series refractive index (RI) detector at 40°C. Samples were loaded in the REZEX ROA (Phenomenex, USA) column ($300 \times 7.8$ mm) at 85°C and eluted with 0.05 M sulfuric acid at a flow rate of 1.5 ml/min.

## Transmission electron microscopy (TEM) analysis of cell wall

Strains were grown statically from $1x10^7$ conidia at 37°C in MM for 24 h. Mycelia were harvested and immediately fixed in 0.1 M sodium phosphate buffer (pH 7.4) containing 2.5% (v/v) of glutaraldehyde and 2% (w/v) of paraformaldehyde for 24 h at 4°C. Samples were encapsulated in agar (2% w/v) and subjected to fixation (1% $OsO_4$), contrasting (1% uranyl acetate), ethanol dehydration, and a two-step infiltration process with Spurr resin (Electron Microscopy Sciences) of 16 h and 3 h at RT. Additional infiltration was provided under vacuum at RT before embedment in BEEM capsules (Electron Microscopy Sciences) and polymerization at 60°C for 72 h. Semithin (0.5-μm) survey sections were stained with toluidine blue to identify the areas of best cell density. Ultrathin sections (60 nm) were prepared and stained again with uranyl acetate (1%) and lead citrate (2%). Transmission electron microscopy (TEM) images were obtained using a Philips CM-120 electron microscope at an acceleration voltage of 120 kV using a MegaView3 camera and iTEM 5.0 software (Olympus Soft Imaging Solutions GmbH). Cell wall thickness of 100 sections of different germlings were measured at 23,500 x magnification and images analyzed with the ImageJ software [77]. Statistical differences were evaluated by using test-T.

## RNA extraction and gene expression analysis

Strains were grown from $1x10^7$ conidia in MM for 20 h at 37°C before being incubated with 200mM $CaCl_2$ for 10 and 30 min. Mycelia were ground to a fine powder in liquid $N_2$ and total RNA was extracted with Trizol reagent (Thermo Scientific) according to the manufacturer's protocol. DNA was digested with Turbo DNase I (Ambion Thermo Scientific) according to manufacturer's instructions. Two μg of total RNA per sample was reverse-transcribed with the High Capacity cDNA Reverse Transcription kit (Thermo Scientifc) using oligo dTV and random primers blend, according to manufacturer's instructions. qRT-PCRs were run in a StepOne Plus Real Time PCR System (Thermo Scientific) and Power Sybr Green PCR Master Mix (Thermo Scientific) was used. Three independent biological replicates were used and the mRNA quantity relative fold change was calculated using standard curves [78]. Primers are described in the S11 Table. RNASeq data (see below) was exploited to select genes for normalization of the qPCR data, applying the method of Yim *et al.* [79]. Briefly, genes with a coefficient of variation smaller than 10%, behaving in a normally distributed manner, and with large expression values (FPKM) were selected. After experimental validation, the gene *cofA* (Afu5g10570) was selected.

## RNA sequencing

*A. fumigatus* conidia ($5x10^7$ sp/ml) from the strains CEA17, Δ*zipD*, and Δ*crzA* were inoculated in triplicate into liquid MM and cultured for 20 h, 37°C prior to addition or not of 200mM $CaCl_2$, for 10 and 30 min. Mycelia were harvested, frozen in liquid nitrogen. For total RNA isolation, mycelia were ground in liquid nitrogen. Total RNA was extracted using Trizol (Invitrogen), treated with RNase-free DNase I (Fermentas) and purified using a RNAeasy Kit (Qiagen) according to manufacturer's instructions. The RNA from each treatment was quantified using a NanoDrop and Qubit fluorometer, and analyzed using an Agilent 2100 Bioanalyzer system to assess the integrity of the RNA. RNA Integrity Number (RIN) was calculated; RNA sample had a RIN = 9.0–9.5.

Illumina TruSeq Stranded mRNA Sample Preparation kit was used. Briefly, polyA containing mRNA molecules were selected using polyT oligo-attached magnetic beads. Fragmentation and paired-end library preparation was done using divalent cations and thermal fragmentation. First strand cDNA synthesis was performed using reverse transcriptase (Superscript II)

and random primers. This was followed by second strand cDNA synthesis using DNA Polymerase I and RNase H and dUTP in place of dTTP. AMPure XP beads were used to separate the dscDNA from the second strand. At the end of this process, we had blunt-ended cDNA to which a single 'A' nucleotide was added at the 3' end to prevent them from ligating to one another during the adapter ligation reaction. A corresponding single 'T' nucleotide on the 3' end of the adapter provided a complementary overhang for ligating the adapter to the fragment. The products were then purified and enriched using a PCR to selectively enrich those DNA fragments that have adapter molecules on both ends and to amplify the amount of DNA in the library. The PCR was performed with a PCR Primer Cocktail from the Illumina kit that anneals to the ends of the adapters. Libraries were sequenced (2x100bp) on CTBE NGS sequencing facility HiSeq 2500 instrument, generating approx. $13.1x10^6$ fragments per sample. Short reads were submitted to the NCBI's Short Read Archive under the Bioproject: PRJNA445394.

Obtained fastq files were quality checked with FastQC (http://www.bioinformatics. babraham.ac.uk/projects/fastqc/), and cleaned (quality trim, adaptor removal and minimum length filtering) with Trimmomatic [80], finally, ribosomal RNA was removed using Sort-MeRNA [81]. High quality reads were mapped to the *A. fumigatus* Af293 genome sequence [82] using Tophat2 [83] in strand-specific mode. Saturation of sequencing effort was assessed by counting the number of detected exon-exon junctions at different sub-sampling levels of the total high-quality reads, using RSeQC [84]. All samples achieved saturation of known exon-exon junctions. Reproducibility among biological replicates was assessed by exploring a principal component analysis (PCA) plot of the top 500 genes have the largest biological variation between the libraries, and by pair-wise measuring the Pearson correlation among the replicates over the whole set of genes. In order to assess transcript abundance exonic reads were counted in a strand-specific way using the featureCounts function from Rsubread Bioconductor package [85]. Calling of differentially expressed genes was carried-out using DESeq2 [86] using as threshold adjusted p-value < 0.01 [87].

## Quantitative reverse transcription (qRT)-PCR

After 6 h of stimulation with UV-inactivated fungal cells, the BMDMs ($2 x 10^6$/ml; $1 x 10^6$ cells/well; 24-well plates) were used to extract the total RNA using the Trizol Reagent (Invitrogen, Life Technologies, Camarillo, CA, USA), according to the manufacturer's instructions. The total RNA was reverse-transcribed into cDNA by the ImProm-II Reverse Transcription System (Promega, Fitchburg, WI, USA) using oligo (dT). qRT-PCR was performed in 15 μl, the reactions were using carried out by SsoFast EvaGreen (Bio-Rad Laboratories, Hercules, CA, USA) on a Bio-Rad CFX96 Real-Time PCR System (Bio-Rad Laboratories) under the following conditions: 95°C for 30 sec, and 40 cycles of 95°C for 15 sec/60°C for 5 sec. The relative expression of transcripts was quantified using ΔΔCt method and normalized to β-actin expression. PCR primers utilized were: β-actin (F-AGCTGCGTTTTACACCCTTT / R-AAGC-CATGCCAATGTTGTCT); T-bet (F-CACTAAGCAAGGACGGCGAA / R-CCACCAAGACCACATCCAC); GATA-3 (F-AAGAAAGGCATGAAGGACGC / R-GTGTGCCCATTTGGACATCA); ROR-γ t (F-TGGAAGATGTGGACTTCGTT / R-TGGTTCCCCAAGTTCAGGAT).

## Murine model of pulmonary aspergillosis and fungal burden

Outbreed female mice (BALB/c strain; body weight, 20 to 22 g) were housed in vented cages containing 5 animals. Mice were immunosuppressed with cyclophosphamide (150 mg per kg of body weight), which was administered intraperitoneally on days -4, -1, and 2 prior to and

post infection. Hydrocortisonacetate (200mg/ kg body weight) was injected subcutaneously on day -3. *A. fumigatus* strains were grown on YAG for 2 days prior to infection. Fresh conidia were harvested in PBS and filtered through a Miracloth (Calbiochem). Conidial suspensions were spun for 5 min at 3,000 x *g*, washed three times with PBS, counted using a hemocytometer, and resuspended at a concentration of $5.0 \times 10^6$ conidia/ ml. The viability of the administered inoculum was determined by incubating a serial dilution of the conidia on YAG medium, at 37˚C. Mice were anesthetized by halothane inhalation and infected by intranasal instillation of $1.0 \times 10^5$ conidia in 20 μl of PBS. As a negative control, a group of 5 mice received PBS only. Mice were weighed every 24 h from the day of infection and visually inspected twice daily. The statistical significance of comparative survival values was calculated by Prism statistical analysis package by using Log-rank (Mantel-Cox) Test and Gehan-Brestow-Wilcoxon tests.

To investigate fungal burden in the lungs, mice were infected as described previously, but with a higher inoculum of $1\times10^6$ conidia/20 μl. A higher inoculum, in comparison to the survival experiments, was used to increase fungal DNA detection. Animals were sacrificed 72 h post-infection, and both lungs were harvested and immediately frozen in liquid nitrogen. Samples were homogenized by vortexing with glass beads for 10 min, and DNA was extracted via the phenol-chloroform method. DNA quantity and quality were assessed using a NanoDrop 2000 spectrophotometer (Thermo Scientific). At least 500 μg of total DNA from each sample was used for quantitative real-time PCRs. Two primers were used to amplify the 18S rRNA region of *A. fumigatus* (primer, 5′-CTTAAATAGCCCGGTCCGCATT-3′; probe, 5′-CATCACAGACCTGTTATTGCCG-3′) and an intronic region of mouse GAPDH (glyceraldehyde-3-phosphate dehydrogenase) (primer, 5′-CGAGGGACTTGGAGGACACAG-3′; probe, 5′-GGGCAAGGCTAAAGGTCAGCG-3′). Six-point standard curves were calculated using serial dilutions of gDNA from all the *A. fumigatus* strains used and the uninfected mouse lung. Fungal and mouse DNA quantities were obtained from the threshold cycle (CT) values from an appropriate standard curve. Fungal burden was determined as the ratio between picograms of fungal and micrograms of mouse DNA.

### *A. fumigatus* infection in immunocompetent mice

Eight- to 12-week-old male C57Bl/7 WT mice were obtained from the specific pathogen free Isogenic Breeding Unit of the Department of Immunology, Institute of Biomedical Sciences, University of São Paulo. Mice were anesthetized and submitted to intra-tracheal (i.t.) infection as previously described [75]. Briefly, after i.p. injection of ketamine and xylazine, animals were infected with $5\times10^7$ CEA17 WT, $\Delta zipD^{CEA17}$ and $\Delta zipD^{CEA17}::zipD^+$ conidia, contained in 75 μl of PBS [88], by surgical i.t. inoculation, which allowed dispensing of the fungal cells directly into the lungs.

### Assessment of leukocyte subpopulations and flow cytometric analysis

Lungs *A. fumigatus*-infected mice were collected after 3 days of infection. To assess the leukocyte subpopulations the lungs were removed and digested enzymatically for 40 minutes with collagenase (2 mg/ml) in RPMI culture medium (Sigma). Total lung leukocyte numbers were assessed with trypan blue, and viability was always >95%. For cell-surface staining, leukocytes were washed and suspended at $1 \times 10^6$ cells/mL in staining buffer (PBS, 2% fetal calf serum and 0.1% NaN$_3$). Fc receptors were blocked by the addition of unlabeled anti-CD16/32 (eBioscience). Leukocytes were then stained in the dark for 25 min at 4˚C with the optimal dilution of each monoclonal antibody. To neutrophils and macrophages: anti-CD11b, Ly6G, F4/80, MHC-II and CD86; lymphocytes: anti-CD4, CD8, CD25 and CD69 (eBiosciences or

BioLegend). Cells were washed twice with staining buffer, fixed with 2% paraformaldehyde (PFA; Sigma). For intracellular detection of cytokines, leukocytes obtained from lungs were stimulated for 6 hours in complete RPMI medium containing 50 ng/mL phorbol 12-myristate 13-acetate, 500 ng/mL ionomycin (Sigma), and 3 mM monensin (eBioscience). Next, cells were labeled for surface molecules (CD4 and CD8) and then treated according to the manufacturer's protocol for intracellular staining using the Cytofix/Cytoperm kit (eBiosciences) and specifics antibodies anti-4, IL-17, and IFN-γ. Cells were washed twice with staining buffer, suspended in 100 μl, and an equal volume of PFA was added to fix the cells. A minimum of 100,000 events was acquired on FACScanto II flow cytometer (BD Biosciences) using the FACSDiva software (BD Biosciences). Lymphocytes, macrophages and neutrophils were gated as judged from forward and side light scatter. The cell surface expression of leukocyte markers as well as intracellular cytokine expression was analyzed using the FlowJo software (Tree Star).

## Bone marrow-derived macrophages (BMDMs) preparation and CFU assay

The BMDMs preparation was performed according to Francke *et al*. [89] with slight modifications. BMDMs were prepared from 8- to 12-week old C57BL/6 adult mice femurs and tibias after flushing with RPMI 1640 medium to release bone marrow cells. These cells were cultured for 5 days in RPMI 1640 medium supplemented with 10% fetal cow serum (FCS) and 20 ng/ml M-CSF (Invitrogen). Non-adherent cells were removed and the adherent cells (macrophages) were removed and washed twice with cold PBS. Cell number was determined using a Neubauer chamber. Two hundred μl of RPMI-FCS containing $1x10^5$ conidia (1:2 macrophage/conidia ratio) was added in 96-well microplates at 37˚C with 5% $CO_2$. After 4h of coculture, plates were centrifuged ($400 \times g$, 10 min, 4˚C), the supernatants discarded, and the pellet maintained in individual tubes. The wells containing adherent cells were washed with distilled water to lyse macrophages and suspensions were collected in the same individual tubes containing the pellet. Thus, the samples contained both ingested and non-ingested conidia. Serial dilutions were assayed on Sabouraud agar plates (Sigma). CFUs were counted after 18 h incubation at 37˚C. All assays were done with four wells per condition in over two independent experiments. For measurement of cytokines, $2x10^5$ conidia (1:5 macrophage/conidia ratio) were cultured in 48-well microplates in 300 uL of RPMI-FCS. The cells were cocultivated for 4 h at 37˚C in 5% $CO_2$ to allow conidia adhesion and ingestion. Cells were washed twice with PBS to remove any non-ingested or non-adhered conidia and the samples were cultured for an addition period of 18 h to permit the cytokine production by adherent cells. The samples were then centrifuged ($400 \times g$, 10 min, 4˚C), supernatants were stored at −70˚C, and further analyzed for the presence of cytokines.

## Cytokines detection

The left lung from *A*. *fumigatus*-infected mice were aseptically removed and individually disrupted in 5 mL of PBS after 3 days of infection. Supernatants were separated from cell debris by centrifugation at 3,000×*g* for 10 min and stored at -80˚C. The levels of IL-1β, IL-4, IL-6, IL-10, IL-12, IL-17, TNF-α, and IFN-γ and were measured by capture enzyme-linked immunosorbent assay (ELISA) with antibody pairs purchased from eBioscience. The levels of IL-1β, IL-6, IL-12, and TNF-α were measured from the supernatants obtained from *in vitro* experiments as described above. Plates were read using a spectrophotometric plate reader (VersaMax, Molecular Devices).

## Histopathology

After been fixed in 10% formaldehyde solution for histopathology methodology, the lungs were diaphanized and embedded in paraffin and then sliced in serial section of about 5 μm thickness. The staining was done with Hematoxylin and Eosin (HE) or GMS (Sigma-Aldrich GMS Kit). The slides were analyzed and the images were recorded using the microscope (Leica DM6000 B).

## Supporting information

**S1 Fig. The distribution ZipD related proteins throughout the fungal tree of life.** BlastP analyses using the mature ZipD protein sequence were performed, at the expected (e-value) cut-off thresholds of 1.0e-5, 1.0e-50 and 1.0e-100, on the predicted proteomes of 791 fungi, representing the 13 different taxonomic classes or subphyla within Dikarya, as presented on the JGI Mycocosm portal. Diagrammatic representation of the fungal tree of life (not indicative of evolutionary time) adapted from the JGI MycoCosm. Pie charts show the number of species genomes within each taxonomic class with at least one protein similar to ZipD at the indicated BlastP thresholds.
(PDF)

**S2 Fig. The wild-type, ΔcrzA, ΔzipD, and ΔcrzA ΔzipD mutant strains were grown for 16 h at 37˚C and transferred to 200 mM CaCl₂ for 0, 10, and 30 mins.** Gene expression was normalized using *cofA* (Afu5g10570). Standard deviations present the average of three independent biological repetitions (each with 2 technical repetitions). Above each graph the corresponding result of the heat-map RNAseq for each gene. Once more, the wild-type is shown as 10 and 30 min calcium stress versus time zero (20 hours growth), and gene deletion strains are shown as the deletion strain versus the equivalent wild-type 10 and 30 min time points (the mutant values have been normalised to the basal level of expression of each gene before stress, i.e., expression ratios are being compared: wild-type 10 min versus time zero divided by a specific mutant 10 and 30 min versus time zero). The expression of these sixteen genes showed a high level of correlation with the RNA-seq data (Pearson correlation from 0.7055 to 0.9187; Fig 4E).
(TIF)

**S3 Fig. PCR scheme to check the ZipD:3xHA strain.** (B) Phenotype analysis of wild type and ZipD:3xHA strains which were grown in MM plates for 5 days at 37˚C.
(PDF)

**S4 Fig. Co-Immunoprecipitation of CalA::GFP and ZipD:3xHA.** (A) PCR scheme to verify the homologous integration of CalA::GFP and CalA::GFP ZipD:3xHA. (B) Phenotypic analysis of wild type, CalA::GFP (candidate 2 in the PCR) and CalA::GFP ZipD:3xHA (candidate 2 in the PCR) strains which were grown in YAG plates, with or without CaCl₂ for 3 days at 37˚C. (C) Verification of interaction between CalA and ZipD by Co-IP. Affinity purification assays from GFP-tagged CalA strain in the background of 3xHA-tagged ZipD were performed with (a) GFP-Trap and (b) anti-HA beads to verify interactions. The coimmunoprecipitated proteins were analysed by the indicated antibodies.
(PDF)

**S5 Fig.** Screening for the phosphatase mutants more sensitive to sorbitol (A), caspofungin (B), and CaCl₂ (C).
(TIF)

**S6 Fig.** (A) The wild-type, ΔzipD, and all phosphatase catalytic subunit null mutants were grown for 16 h at 37˚C and transferred to 200 mM $CaCl_2$ for 0 and 10 mins. Gene expression was normalized using cofA (Afu5g10570). Standard deviations present the average of three independent biological repetitions (each with 2 technical repetitions). Statistical analysis was performed using a one-way ANOVA test when compared to the wild-type condition (*$p<0.05$). (B) The wild-type, ΔzipD, and four conditional were grown for 16 h at 37˚C in MM +nitrate as a single nitrogen source, and then transferred to MM+ammonium tartrate as a single nitrogen source, and subsequently to 200 mM $CaCl_2$ for 0 and 10 mins. Gene expression was normalized using cofA (Afu5g10570). Standard deviations present the average of three independent biological repetitions (each with 2 technical repetitions). Statistical analysis was performed using a one-way ANOVA test when compared to the wild-type condition (*$p<0.05$).
(TIF)

**S1 Table. List of *Aspergillus fumigatus* genes encoding transcription factors deleted.**
(XLS)

**S2 Table. Genes that displayed about the same expression levels at the different treatments.**
(XLSX)

**S3 Table. Genes transcriptionally modulated comparing the wild-type 10 min with the control.**
(XLSX)

**S4 Table. Genes transcriptionally modulated comparing the wild-type 30 min with the control.**
(XLSX)

**S5 Table. Genes transcriptionally modulated comparing the Δ*zipD* with the wild-type 10 min.**
(XLSX)

**S6 Table. Genes transcriptionally modulated comparing the Δ*zipD* with the wild-type 30 min.**
(XLSX)

**S7 Table. Genes transcriptionally modulated comparing the Δ*crzA* with the wild-type 10 min.**
(XLSX)

**S8 Table. Genes transcriptionally modulated comparing the Δ*crzA* with the wild-type 30 min.**
(XLSX)

**S9 Table. Diameters of the cell walls of the wild-type and mutant strains.**
(XLSX)

**S10 Table. *Aspergillus fumigatus* phosphatase mutants.**
(DOCX)

**S11 Table. List of primers used in this work.**
(XLSX)

## Author Contributions

**Conceptualization:** Gustavo H. Goldman.

**Data curation:** Gabriela Felix Persinoti.

**Formal analysis:** Patrícia Alves de Castro, Ana Cristina Colabardini, Adriana Oliveira Manfiolli, Eliciane Cevolani Mattos, Giuseppe Palmisano, Fausto Almeida, Laure Nicolas Annick Ries, Laura Mellado, Marina Campos Rocha, Roberto Nascimento Silva, Flávio Vieira Loures, Iran Malavazi.

**Funding acquisition:** Gustavo H. Goldman.

**Investigation:** Patrícia Alves de Castro, Ana Cristina Colabardini, Adriana Oliveira Manfiolli, Jéssica Chiaratto, Lilian Pereira Silva, Eliciane Cevolani Mattos, Giuseppe Palmisano, Fausto Almeida, Laure Nicolas Annick Ries, Laura Mellado, Marina Campos Rocha, Roberto Nascimento Silva, Gabriel Scalini de Souza, Flávio Vieira Loures, Iran Malavazi.

**Project administration:** Gustavo H. Goldman.

**Resources:** Michael Bromley.

**Supervision:** Gustavo H. Goldman.

**Writing – original draft:** Neil Andrew Brown, Gustavo H. Goldman.

**Writing – review & editing:** Neil Andrew Brown, Gustavo H. Goldman.

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
