## [Decision Letter · Decision Letter 0]

30 Oct 2019

Dear Dr Goldman,

Thank you very much for submitting your Research Article entitled 'Aspergillus fumigatus calcium-responsive transcription factors regulate cell wall architecture promoting stress tolerance, virulence and caspofungin resistance' to PLOS Genetics.

The manuscript was fully evaluated at the editorial level and by the same peer reviewers as the previous submission. As you will see, all three reviewers are generally positive, but there are some important concerns that will need to be addressed for the manuscript to move forward.

We therefore ask you to modify the manuscript according to the review recommendations before we can consider your manuscript for acceptance. Your revisions should address the specific points made by each reviewer. In addition we ask that you provide a detailed list of your responses to the review comments and a description of the changes you have made in the manuscript.

We hope to receive your revised manuscript within the next 60 days. If you anticipate any delay in its return, we would ask you to let us know the expected resubmission date by email to plosgenetics@plos.org.

[LINK]

Yours sincerely,

Gregory S. Barsh

Editor-in-Chief

PLOS Genetics

Gregory Copenhaver

Editor-in-Chief

PLOS Genetics

Reviewer's Responses to Questions

**Comments to the Authors:**

Reviewer #1: In this manuscript by Alves de Castro et al., the authors characterized a novel transcription factor (ZipD) in Aspergillus fumigatus. The authors used various approaches and presented the meaningful results in this manuscript. The authors concluded that ZipD plays important roles in cell wall organization, osmotic stress response and virulence. Overall, this manuscript can provide meaningful results in the fungal biology field. However, this manuscript should need to be revised before it is published.

1. First of all, it seems that the authors did not confirm properly before submitting the manuscript. There are missing Figure 7. Without Figure 7, this manuscript cannot be reviewed.

2. The authors should check Figure 3 and its explanation. It’s not match at all.

3. The calcium concentrations used in the experiments in this manuscript are different. Mammalian environment has roughly 2mM calcium concentration, but the authors used 200 mM calcium for RNA-seq and localization, and 10 mM Calcium for proteomic analysis.

4. The zipD deletion mutant is more sensitive to caspofungin and cell wall damaging agents, but not the crzA deletion mutant. How to explain this situation. Can RNA-seq results or other results explain this phenomenon? It can be discussed.

Minor comments

Line 167, crzA italic

Line 178 replaced “Figure 1A” with “Figure 1B)

Line 190, Afu5g10620 is also involved in the calcium-calcineurin pathway and tolerance.

Line 217-221, How about the vcxB gene expression in zipD crzA double deletion mutant?

Line 238-240, The authors did not check the growth inhibition in 200 mM calcium concentration.

Line 235-254, The authors simply lists the number of genes affected zipD and crzA deletion mutants. However, it is doubtful whether this analysis provides accurate knowledge. It’s more useful that the authors should check the pattern of mRNA expression based on calcium exposure times. For example, the authors explained that 516 genes (total 588 in both 10 and 30 min) were up-regulated genes in both mutants. In other words, only 72 genes were upregulated in both 10 and 30 min. The RNA-seq results can be needed to analyze in more detail.

Reviewer #2: In this second revised manuscript, the authors provided several additional data sets to further elucidate the regulatory mechanism of ZipD in association with calcium-dependent calcineurin signaling pathway. First, they screened the Aspergillus fumigatus transcription factor mutant library and identified several other TFs, besides ZipD and CrzA, involved in calcium signaling and caspofungin-paradoxical effects (CPE). Second, the authors demonstrated that phosphorylation status determines nuclear localization of ZipD. This was a very nice addition, as it was one of this reviewer’s main questions. Finally, they performed additional experiments with the non-essential phosphatase null mutants, and essential phosphatase conditional mutant libraries and demonstrated that ZipD can be dephosphorylated by multiple phosphatases.

Overall these new dataset significantly improved the quality of this manuscript and provide more mechanistic insight into the ZipD transcription factor in A. fumigatus. I have only the following minor comment.

1. Figure 6E should be presented as graphs along with actual cellular localization images of ZipD-GFP and its phosphomutants. Alternatively, Figure 6E-G could be presented as independent figure.

Reviewer #3: In this revised manuscript comparative transcriptomics of nine transcription factors important to calcium stress tolerance revealed that CrzA and ZipD regulate the expression of shared and unique gene networks. Both promoted calcium stress tolerance, but ZipD also regulated cell wall organization, osmotic stress tolerance and echinocandin resistance. Based on a series of complementary experiments, the results presented support the conclusion that ZipD has unique transcriptional signatures that regulate cell wall composition and the osmotic stress response. The ZipD-mediated regulation of the cell wall has important implications in virulence and anti-fungal drug resistance – both conclusions are supported by robust experimentation and in vivo analysis. The current study represents a revised manuscript that was previously reviewed. The authors were responsive to the comments and concerns that were raised by other reviewers. As a result the present manuscript is improved given that the most serious concerns were addressed. In addition, the manuscript is clearly-written and it addresses an interesting and relevant aspect of the biology and virulence of Aspergillus fumigatus. This is a medically-relevant pathogen and as with so many of the most harmful fungi, we still lack a clear and comprehensive understanding of fungal physiology, and mechanisms of virulence and pathogenesis. This study makes a significant contribution and it underscores the complexity and interplay between mechanisms mediating calcium signaling, fungal cell stress response, cell wall composition and fungal disease.

**Have all data underlying the figures and results presented in the manuscript been provided?**

Reviewer #1: No: There is no Figure 7.

Reviewer #2: Yes

Reviewer #3: Yes

PLOS authors have the option to publish the peer review history of their article (what does this mean?). If published, this will include your full peer review and any attached files.

Reviewer #1: No

Reviewer #2: No

Reviewer #3: No

---

## [Decision Letter · Decision Letter 1]

2 Dec 2019

Dear Dr Goldman,

We are pleased to inform you that your manuscript entitled "Aspergillus fumigatus calcium-responsive transcription factors regulate cell wall architecture promoting stress tolerance, virulence and caspofungin resistance" has been editorially accepted for publication in PLOS Genetics. Congratulations!

Please note the comments from reviewer #1, and address those concerns during the production process.

Yours sincerely,

Gregory S. Barsh

Editor-in-Chief

PLOS Genetics

Gregory Copenhaver

Editor-in-Chief

PLOS Genetics

Comments from the reviewers (if applicable):

Reviewer's Responses to Questions

Comments to the Authors:

Please note here if the review is uploaded as an attachment.

Reviewer #1: In this revised manuscript, the authors addressed most issues raised by reviewers. It can be published to PLoS Genetics.

Please check lines 497 and 499. Figure 7B should be revised.

Reviewer #2: The authors addressed my comments nicely. I don't have any more comments.

Have all data underlying the figures and results presented in the manuscript been provided?

Large-scale datasets should be made available via a public repository as described in the 

PLOS Genetics

data availability policy, and numerical data that underlies graphs or summary statistics should be provided in spreadsheet form as supporting information.

Reviewer #1: Yes

Reviewer #2: Yes

PLOS authors have the option to publish the peer review history of their article (what does this mean?). If published, this will include your full peer review and any attached files.

Do you want your identity to be public for this peer review?

 For information about this choice, including consent withdrawal, please see our Privacy Policy.

Reviewer #1: No

Reviewer #2: No

**Data Deposition**

http://datadryad.org/submit?journalID=pgenetics&manu=PGENETICS-D-19-01643R1

Press Queries

---

## [Editor Report · Acceptance letter]

20 Dec 2019

PGENETICS-D-19-01643R1 

*Aspergillus fumigatus* calcium-responsive transcription factors regulate cell wall architecture promoting stress tolerance, virulence and caspofungin resistance 

Dear Dr Goldman, 

We are pleased to inform you that your manuscript entitled "*Aspergillus fumigatus* calcium-responsive transcription factors regulate cell wall architecture promoting stress tolerance, virulence and caspofungin resistance" has been formally accepted for publication in PLOS Genetics! Your manuscript is now with our production department and you will be notified of the publication date in due course.

With kind regards,

Nicholas White

PLOS Genetics

On behalf of:
